# ValUES: A Framework for Systematic Validation of Uncertainty Estimation in Semantic Segmentation

**Kim-Celine Kahl**[1,2*]**, Carsten T. Lüth**[1,2*]**, Maximilian Zenk**[3]**,**
**Klaus Maier-Hein**[2,3]**, Paul F. Jaeger**[1,2]

[1]German Cancer Research Center (DKFZ) Heidelberg, Interactive Machine Learning Group, Germany
[2]Helmholtz Imaging, German Cancer Research Center (DKFZ), Heidelberg, Germany
[3]German Cancer Research Center (DKFZ) Heidelberg, Division of Medical Image Computing, Germany

`{k.kahl, carsten.lueth}@dkfz-heidelberg.de`

## Abstract

Uncertainty estimation is an essential and heavily-studied component for the reliable application of semantic segmentation methods. While various studies exist claiming methodological advances on the one hand, and successful application on the other hand, the field is currently hampered by a gap between theory and practice leaving fundamental questions unanswered: Can data-related and model-related uncertainty really be separated in practice? Which components of an uncertainty method are essential for real-world performance? Which uncertainty method works well for which application? In this work, we link this research gap to a lack of systematic and comprehensive evaluation of uncertainty methods. Specifically, we identify three key pitfalls in current literature and present an evaluation framework that bridges the research gap by providing 1) a controlled environment for studying data ambiguities as well as distribution shifts, 2) systematic ablations of relevant method components, and 3) test-beds for the five predominant uncertainty applications: OoD-detection, active learning, failure detection, calibration, and ambiguity modeling. Empirical results on simulated as well as real-world data demonstrate how the proposed framework is able to answer the predominant questions in the field revealing for instance that 1) separation of uncertainty types works on simulated data but does not necessarily translate to real-world data, 2) aggregation of scores is a crucial but currently neglected component of uncertainty methods, 3) While ensembles are performing most robustly across the different downstream tasks and settings, test-time augmentation often constitutes a light-weight alternative. Code is at: `https://github.com/IML-DKFZ/values`

## 1 Introduction

In order to reliably deploy image segmentation systems in real-world applications, there is a critical need to estimate and quantify the uncertainty associated with their predictions. Despite numerous studies on uncertainty methods for segmentation in recent years, their effective utilization is currently hindered by a significant gap between the theoretical development and their application in relevant downstream tasks. One aspect of this disparity is the fact that uncertainty methods are often stated to model a specific type of uncertainty, i.e., either the data-related, aleatoric uncertainty (AU) or the model-related, epistemic uncertainty (EU). However, explicit evaluation of the claimed behavior is not the focal point of these studies. As a prominent example, two highly-cited studies built on the claim that test-time data augmentations (TTA) improve a model's ability to capture AU, without theoretical or empirical evidence for this hypothesis (Wang et al. (2019); Ayhan & Berens (2018)).

---

[*]These authors contributed equally to this work

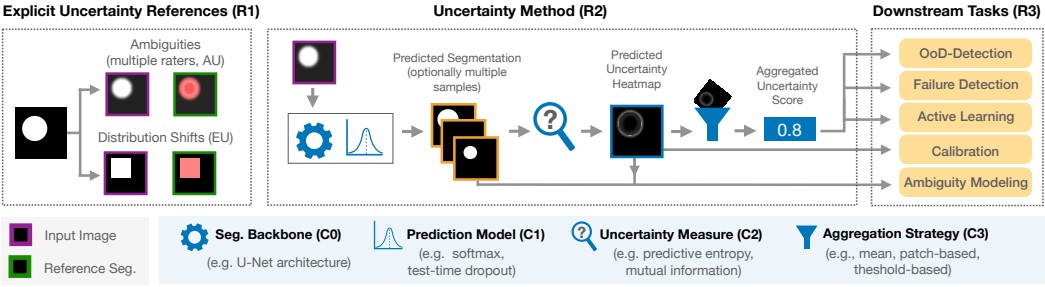

Figure 1: **Framework for systematic validation of uncertainty methods in segmentation**. With our framework, we aim to overcome pitfalls in the current validation of uncertainty methods for semantic segmentation by satisfying the three requirements (R1-R3) for a systematic validation: We explicitly control for aleatoric and epistemic uncertainty in the data and references (R1). We define and validate four individual components C0-C3 of uncertainty methods (R2): First, one or multiple segmentation outputs are generated by the segmentation backbone (C0) and the prediction model (C1). Next, an uncertainty measure is applied (C2) producing an uncertainty heatmap, which can be aggregated using an aggregation strategy (C3). Finally, the real-world capabilities of methods need to be validated on various downstream tasks (R3).

However, a follow-up study proposed the opposite, i.e., TTA to model EU, without validating this statement either (Hu et al. (2019)). Thus, there is a clear need to 1) validate the stated behavior of methods regarding feasibility of separation and 2) provide evidence for the necessity of separation by checking whether applications actually benefit. Another aspect of the current research gap is the underexplored study of all practically relevant components of an uncertainty method. For instance, an adequate aggregation of uncertainty estimates from pixel level to image level is highly relevant to performance in many downstream tasks but often overlooked, leading to tasks like failure detection being purely validated on pixel-level instead of image-level (Zhang et al. (2022); Mehta et al. (2020)) or simplistic aggregation strategies being employed (Gonzalez et al. (2021); Czolbe et al. (2021)). Finally, the current gap between theory and application is nurtured by the fact that proposed methods are rarely validated on a broad set of relevant downstream tasks, making it difficult and expensive for practitioners to identify the best uncertainty method for their problem.

This work bridges the gap between theoretical advancements in uncertainty estimation and its real-world application in segmentation systems by presenting a framework for standardized and systematic validation (see Figure 1). The framework features 1) a controlled environment for studying data ambiguities as well as distribution shifts, 2) systematic ablations of relevant method components, and 3) test-beds for the five predominant uncertainty applications: Out-of-Distribution-detection (**OoD-D**), active learning (**AL**), failure detection (**FD**), calibration (**CALIB**), and ambiguity modeling (**AM**). We demonstrate the effectiveness of our proposed framework based on an exemplary empirical study that sheds light on the unanswered questions and current inconsistencies in the field and allows us to compile a list of hands-on recommendations.

## 2    UNCERTAINTY ESTIMATION IN SEMANTIC SEGMENTATION

To effectively discuss pitfalls and challenges in the field of uncertainty estimation for segmentation, we begin by establishing a common language by defining components of an uncertainty method.

**C0 - Segmentation Backbone.** The segmentation backbone is the fundamental building block for uncertainty estimation, depicting the method's architecture, e.g., a U-Net architecture (Ronneberger et al. (2015)). As well-established architectures exist, C0 is often fixed in uncertainty studies.

**C1 - Prediction Model.** The prediction model (PM) operates based on the segmentation backbone and produces the final predicted class scores for segmentation. Depending on the PM, a single set ("deterministic") or multiple sets ("sampling-based") of scores per input image can be generated. The PM may include dedicated training and inference paradigms, like ensemble training or test-time dropout (TTD). Examples of PMs include deterministic models like softmax, Bayesian approaches, and probabilistic models like stochastic segmentation networks (SSNs) (Monteiro et al. (2020)).

**C2 - Uncertainty Measure.** The uncertainty measure involves computing an uncertainty score per pixel based on predicted class scores, which can be represented as an uncertainty heatmap. Examples of uncertainty measures include expected entropy and mutual information.

**C3 - Aggregation Strategy.** The aggregation strategy is a unique component of uncertainty estimation for semantic segmentation that is not needed in tasks such as image classification. Here, the pixel-level uncertainty heatmap is aggregated to a single scalar value at the desired level of granularity depending on the downstream task (e.g., patch-level or image-level). A simple example of an aggregation strategy is to compute the sum or mean over the pixel-level uncertainties.

## 2.1 Measuring Uncertainties

In literature, typically, two types of uncertainty are distinguished: aleatoric uncertainty (AU) and epistemic uncertainty (EU) (Kendall & Gal (2017)). AU relates to inherent ambiguities in the image, such as those caused by spatial occlusions. On the other hand, EU relates to the model itself and arises from a lack of knowledge, which can be mitigated by incorporating additional relevant knowledge, such as images, into the training data. The combination of AU and EU is referred to as predictive uncertainty (PU). The most prominent approach to capture these uncertainties was introduced by (Kendall & Gal (2017)), viewing it from the perspective of a Bayesian classifier, which receives an input $x$ and outputs the probabilities for classes $Y$: $p(Y|x) = \mathbb{E}_{\omega \sim \Omega}[p(Y|x, \omega)]$, where the model parameters $\Omega$ follow $p(\omega|\mathcal{D})$ given the training data $\mathcal{D}$. This Bayesian framework (Mukhoti et al. (2021)) assumes the predictive entropy (PE) to represent the PU which is the sum of the mutual information (MI) representing the EU and the expected entropy (EE) representing the AU:

$$\underbrace{H(Y|x)}_{\text{PU}} = \underbrace{\text{MI}(Y, \Omega|x)}_{\text{EU}} + \underbrace{\mathbb{E}_{\omega \sim \Omega}[H(Y|\omega, x)]}_{\text{AU (for i.i.d. } x)} \tag{1}$$

Here $H$ stands for Shannon's entropy (Shannon (1948)). Besides this, alternative functions like density estimators, such as the Mahalanobis distance to the i.i.d. (independent and identically distributed) training distribution, can be used to approximate uncertainty (Gonzalez et al. (2021)).

## 3 Pitfalls and solutions for a systematic validation of uncertainty methods

Our goal is to bridge the gap between theory and practical application of uncertainty methods in segmentation. To this end, we formulate three requirements (R1-R3) for evaluation protocols aiming to deepen the understanding of how uncertainty methods behave in application and thus allow a safe and reliable deployment of segmentation systems. For each requirement, we also make the described gap explicit by stating the pitfalls of current validation practices in the field.

**R1: Evaluate uncertainty methods claiming to separate AU and EU by means of explicit references and metrics.** Theoretical studies often make claims about a specific uncertainty method capturing either EU or AU. Validating this claimed behavior requires 1) for AU a test set with references from multiple raters that reflect the ambiguities in the data and a metric that explicitly assesses the capturing of these ambiguities such as the normalized cross-correlation (NCC) (Hu et al. (2019)), and 2) for EU a test set featuring samples with explicit distribution shift (i.e. induced EU) and a metric that explicitly assesses whether an EU-measure can separate these cases, such as the Area Under the Receiver Operating Characteristic Curve (AUROC). As described in the pitfall below, the current state of research lacks a systematic validation of uncertainty modeling, leaving fundamental questions unanswered: Can AU and EU be separated in practice? To what extent can different applications benefit from a potential separation? We design a specific study to answer the open questions regarding the separation of EU and AU in simulated and real-world settings.

**Pitfalls of current practice:** Several AU studies feature test sets with only a single rater (Wang et al. (2019); Whitbread & Jenkinson (2022); Kendall & Gal (2017)) whilst in EU-studies no distribution shifts are used for evaluation (Mukhoti & Gal (2018); Mobiny et al. (2021); Whitbread & Jenkinson (2022)). Further, current studies commonly do not report the required metrics but either segmentation performance (Zhang et al. (2022)), CALIB (Wang et al. (2019); Postels et al. (2019)), FD (Zhang et al. (2022); Mukhoti et al. (2021); Mobiny et al. (2021)), or are based on visual inspection (Mukhoti et al. (2021); Wang et al. (2019); Whitbread & Jenkinson (2022); Mobiny et al. (2021)). In

the prominent study by (Kendall & Gal (2017)), an explicit validation of EU on a distribution shift is performed; however, only comparing raw EU-scores over data sets instead of assessing the separation power with AUROC and using predictive entropy as an EU-measure, thereby contradicting Equation 1. As a consequence of these pitfalls, confusion and contradictions arise, such as the fact that different studies claim TTA to either specifically capture EU (Hu et al. (2019)) or AU (Ayhan & Berens (2018); Wang et al. (2019)) without providing quantitative evidence for their claim.

**R2: Evaluate uncertainty methods with regard to all components of an uncertainty method.** In order to assess the capabilities of an uncertainty method, it is crucial to trace back improvements to its individual components C0-C3 (see Sec. 2) and, in the case of a proposed variation of one component, study how this interacts with the others. Only such rigorous analysis allows identifying scientific progress and fosters a deeper understanding of uncertainty estimation in segmentation.

**Pitfalls of current practice:** A common pattern in current literature is to focus on a potential improvement in one component without attending to the others, such as in the form of a single simplified setting. For instance, Gonzalez et al. (2021) study a specific uncertainty measure (C2) that does not require aggregation while applying a simple "mean aggregation" to all baselines, which can be heavily affected by the number of foreground pixels. This leaves it unclear whether the reported improvement comes from the proposed C2 or, in fact, from the subpar aggregation strategy of baselines (C3). Similarly, Czolbe et al. (2021) use a "sum aggregation" for AL, which might result in querying larger objects. Another common pattern is that studies report only pixel-level downstream tasks and neglect image-level tasks that would require aggregation. Examples are studies reporting only CALIB (Wang et al. (2019); Gustafsson et al. (2020); Hu et al. (2019); Postels et al. (2021)) or pixel-level FD (Zhang et al. (2022); Mehta et al. (2020)). However, the task of FD aims to identify and defer faulty subjects or inputs for e.g. human analysis, which questions a plausible application for deferring individual pixels. In contrast, Jungo et al. (2020) follow R2 by studying and ablating individual components of uncertainty methods. Despite this exception, we argue a general reflection by the community on validation practice in this context is required to overcome this pitfall at scale.

**R3: Evaluate uncertainty methods on all relevant downstream tasks.** Next to theoretical studies and claims of separating uncertainty types, it is important to state that uncertainty estimation is no self-purpose. Instead, it needs to come with a clearly stated purpose, which has to be validated on real-life applications. In order for practitioners to decide whether an existing uncertainty method is adequate for their specific task, it is crucial that proposed methods are generally validated on a broad spectrum of downstream tasks such as OoD-D, AL, FD, CALIB, and AM.

**Pitfalls of current practice:** In current literature, most studies validate uncertainty methods on a single downstream task such as OoD-D (Lambert et al. (2022); Holder & Shafique (2021)), FD (Zhang et al. (2022); Mukhoti et al. (2021); Mobiny et al. (2021)), AL (Mackowiak et al. (2018); Colling et al. (2020); Xie et al. (2020)), CALIB (Wang et al. (2019); Gustafsson et al. (2020); Hu et al. (2019); Postels et al. (2021); Mehrtash et al. (2020)), or AM (Kohl et al. (2018); Monteiro et al. (2020)). Also, some task formulations are limited in scope, such as FD purely on i.i.d. test data not considering failure sources from potential distribution shifts, a pitfall that has been studied recently for classification tasks Jaeger et al. (2023). However, the more general pitfall in this context is that the underlying concepts of a proposed uncertainty method are typically not bound to a single application, but studying their general usability on a broad set of downstream tasks is relevant to the community. Thus, the current practice of sparse task validation poses a major challenge to practitioners who seek to choose the best method for their particular problem.

## 4 EMPIRICAL STUDY

### 4.1 STUDY DESIGN

**Uncertainty separation study.** In this comprehensive separation study, our primary focus lies in investigating the ability of uncertainty measures to effectively separate AU and EU, a claim commonly made in theoretical works (Kendall & Gal (2017)). With the understanding that uncertainty measures are often associated with specific uncertainty types (see Equation 1), we test for different prediction models (C1) to see whether the corresponding uncertainty measures (C2) successfully capture their theoretically claimed uncertainty types. We express the task of separating AU and EU by formulating four questions:

**Q1** Do AU-measures capture AU?     **Q2** Do EU-measures capture AU?
**Q3** Do EU-measures capture EU?     **Q4** Do AU-measures capture EU?

*Q1 + Q2:* For assessing the capability of uncertainty methods in capturing AU, we employ the normalized cross-correlation (NCC) as a quantitative measure between the predicted uncertainty map and the reference uncertainty map based on the disagreement of multiple raters (for details see Appendix A). Additionally, we perform qualitative inspections of the uncertainty maps. Based on the theory, successful separation of uncertainties would imply AU-measures to exhibit high NCC and high qualitative fidelity (Q1 = "yes") and vice versa for EU-measures (Q2 = "no").

*Q3 + Q4:* To evaluate the capability of uncertainty methods in capturing EU, we measure the performance of separating cases of an induced distribution shift (associated with EU) from the i.i.d. cases by means of the AUROC ranking metric on image level. Since the expected spatial manifestation of epistemic uncertainty in the image is not known, we do not perform qualitative inspection for this task. Based on the theory, successful separation of uncertainties would imply EU-measures to exhibit high AUROC (Q3 = "yes") and AU-measures to exhibit low AUROC (Q4 = "no").

We conduct this separation study on different datasets, including a toy dataset, the LIDC-IDRI (LIDC) dataset with two metadata shifts, and the GTA5/Cityscapes (GTA5/CS) dataset. Sec. 4.2 provides detailed information on the specific data set settings.

**Evaluation on downstream tasks.** Through this study, we aim to comprehensively understand the performance and capabilities of various uncertainty methods in practical settings. The study is performed on the LIDC dataset, again with two metadata shifts, and the GTA5/CS dataset. Concretely, we evaluate the following downstream tasks (see Appendix A for task definitions and metric details): **1) OoD-D**, where the goal is to identify images that exhibit distribution shifts from the training data, and measured by means of the AUROC. **2) FD**, where the focus is on identifying images on which the overall segmentation is unsatisfactory based on the Dice score as measured by the AURC (Jaeger et al. (2023)). To further provide a ranking of uncertainty methods independent of the segmentation performance, we adapt the E-AURC (Geifman et al. (2019)) for the segmentation task. **3) AL**, where evaluation is performed at the image level, aiming to select the most informative and representative images from an unlabeled pool to improve the model's performance. To measure the performance of uncertainty methods in querying informative samples, we calculate the relative improvement of the Dice score between a first training (starting budget) and second training (including the queried samples) and subtract the performance increase of random querying. **4) CALIB**, where we measure the Average Calibration Error (ACE) (Neumann et al. (2018); Jungo et al. (2020)) in combination with Platt scaling following Jaeger et al. (2023) to assess the model's reliability of uncertainty estimates at pixel level. **5) AM**, where we assess uncertainty methods' ability to model and quantify label ambiguity at the pixel level. This includes measuring the NCC (Hu et al. (2019)) as well as the Generalized Energy Difference (GED) (Kohl et al. (2018)) between segmentation outputs and reference segmentations to evaluate sample diversity.

## 4.2 UTILIZED DATASETS

The following section provides an overview of all datasets with the most important information, especially how we meet R1 by inducing both AU and EU. For more details on datasets, we refer to Appendix B. Both real-world datasets are split into i.i.d. and OoD sets. The initial models are only trained on the i.i.d. sets and evaluated on i.i.d. and OoD sets for the respective downstream tasks.

**Toy dataset** We generate a 3D toy dataset comprising spheres and cubes as target structures for segmentation. To simulate AU, we add Gaussian blur to the border of the spheres and simulate three raters to provide different segmentation styles at the border. EU is simulated by introducing distribution shifts in the geometric objects, such as changes in color, shape, and position, along with background noise to prevent shortcut learning (Geirhos et al. (2020)). As the toy dataset is designed to answer the questions posed in the separation study (see Sec. 4.1), we created the following training and testing scenarios:

1. Q1+Q2: Training models on data with induced AU; testing on i.i.d. data also containing AU
2. Q3 + Q4: Training models on data without ambiguity; testing on i.i.d. data and shifted data
3. Q4: Training models on data with AU; testing on (a) i.i.d. data and shifted data without AU and (b) i.i.d. data with AU (blur) and shifted data without AU (blur) (see Sec. B.1.1 for details)

**LIDC-IDRI (LIDC)** To study uncertainty methods in a real-world setting, we use the LIDC-IDRI dataset (Armato III et al. (2011)), with the task to segment long nodules in 64x64x64 crops from 3D CT volumes, similarly to Kohl et al. (2018). We only include nodules that have been annotated by four different raters serving as an AU reference. We further induce EU by designing two distribution shifts based on two metadata features, which are textured (i.i.d) vs. non-textured (OoD) (texture shift/ LIDC TEX) and benign (i.i.d) vs. malignant (OoD) nodules (malignancy shift/ LIDC MAL).

**GTA5/Cityscapes (GTA5/CS).** We use the GTA5 (Richter et al. (2016)) and Cityscapes (Cordts et al. (2016)) datasets jointly, with both datasets being comprised of the same classes. To induce AU, we employ a similar strategy as Kohl et al. (2018): We randomly flip some classes ("sidewalk", "person", "car", "vegetation" and "road") with a probability of $\frac{1}{3}$ from <class> to <class 2>. We simulate EU with the shift from GTA5 (i.i.d.) to Cityscapes (OoD).

### 4.3 STUDIED UNCERTAINTY METHODS

**C0: Segmentation Backbone.** For the toy and the LIDC datasets, we use the 3D U-Net architecture as the segmentation backbone (Ronneberger et al. (2015)), as this is a simple and well-established architecture in medical image segmentation. For the GTA5/CS dataset, we use the HRNet (Wang et al. (2020)), which has been shown to achieve state-of-the-art results on the Cityscapes dataset. We keep C0 fixed but it can be varied for future experiments. For implementation details, see Sec. C.1.

**C1: Prediction Model** We study five different prediction models: A plain deterministic softmax model, a model using MC-Dropout at test-time (TTD), and an ensemble of 5 models both viewed from a Bayesian perspective, a softmax model using data augmentations at test-time (TTA) and a Stochastic Segmentation Network (SSN). The SSN is a specific type of model that directly learns to predict AU by producing multiple plausible segmentations for a given input by using a random variable that represents the variability of segmentations. For implementation details, see Sec. C.2.

**C2: Uncertainty Measure** We validate various uncertainty measures stated to capture either the PU, EU, or AU. For the deterministic model, we study the maximum softmax response (MSR) as a measure of PU by calculating $1 - \text{MSR}$. For the Bayesian models, we study the predictive entropy as a measure for PU, $\text{MI}(Y, \omega|x)$ as a measure for EU, and the expected entropy as a measure for AU (see Sec. 2.1). For the TTA model, introducing the random augmentation variable $T$, we assume that the predictive entropy is a measure for PU, $\text{MI}(Y, T|x)$ as a measure for EU, and the expected entropy as a measure for AU (see Appendix E). For the SSN, which uses a variable to model the variability of the label maps, we assume that the predictive entropy measures PU, the expected entropy measures EU, and $\text{MI}(Y, Z|x)$ with the variable $Z$ is a measure for AU (see Appendix D).

**C3: Aggregation Strategy** We validate three different aggregation strategies, taking the pixel-level uncertainties as input and returning one score per image: 1) *Image level aggregation*, as used in Czolbe et al. (2021); Gonzalez et al. (2021); Jungo et al. (2020), where uncertainty scores for all pixels are summed per image. We find that for segmentation maps with one single object in the foreground, the uncertainty score directly correlates with the size of the target object (see Appendix F). Thus, we only use this aggregation strategy on the GTA5/CS dataset. 2) *Patch level aggregation*, which uses a sliding window of size $10^D$ ($D$ is the dimensionality of the image) to sum the uncertainties inside the window, and then selects the patch with the highest uncertainty as the image-level uncertainty score. 3) *Threshold level aggregation* considers only uncertainty scores above a threshold $\lambda$ (see Appendix F for how this threshold is determined) as uncertain, and calculates the mean of those scores. Notably, as selecting the threshold depends on the foreground object size, this strategy is not applicable to the GTA5/CS dataset.

### 4.4 RESULTS OF THE SEPARATION STUDY

The general findings of the uncertainty separation study are summarized in Figure 2a, while underlying results are shown in Figure 2b for the toy dataset and Figure 3 (see gray-shaded "Q" indicators on respective panels) for LIDC and GTA5/CS. More detailed descriptions and results as well as a qualitative analysis of uncertainty maps are provided in Appendix G.

**Modeling aleatoric uncertainty (Q1 + Q2)** While AU-measures clearly captured AU much better than EU-measures for the toy dataset, this behavior is inconsistent on the real-world datasets. On the LIDC datasets with AU stemming from rater ambiguity, which mostly occurs at the border of

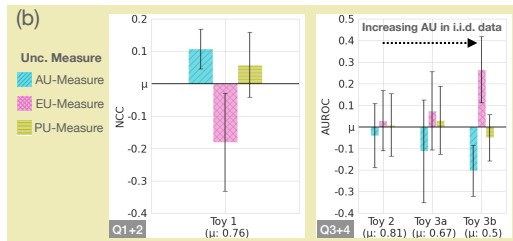

| | Question | Qual./Quant. | Toy Dataset | LIDC | GTA5/CS |
|---|---|---|---|---|---|
| Q1: Do AU-measures capture AU? | | Quantitative | yes | partially | partially |
| | | Qualitative | yes | often (i.i.d.) | partially |
| Q2: Do EU-measures capture AU? | | Quantitativ | no | often yes | no |
| | | Qualitative | no | often yes | no |
| Q3: Do EU-measures capture EU? | | Quantitativ | yes | yes | yes |
| Q4: Do AU-measures capture EU? | | Quantitativ | depends on AU in data | partially | mostly not |

Figure 2: **a)** General findings of the separation study. Green/red denotes agreement/disagreement with theoretical claims, orange represents partial agreement. **b)** Underlying quantitative results on the toy data set. Results show C2 and are aggregated over C1 and C3. Results for LIDC and GTA5/CS are displayed in Figure 3 (see gray-shaded "Q" indicators). Details are in Appendix G
.

structures, the benefit of separating AU-measures from EU-measures is not evident when examining the NCC scores. For GTA5/CS, where induced label ambiguities span entire spatial structures, the AU-measures generally capture AU better than EU-measures. However, the absolute NCC scores from the AU-measures vary greatly across prediction models. We attribute this to SSNs capturing the widespread label ambiguities, while other models overemphasize the border regions.

**Modeling epistemic uncertainty (Q3 + Q4)** While EU-measures capture EU better than AU- and PU-measures on all datasets, the benefit of this separation varied greatly depending on the AU in the respective training and test data. More specifically, when more AU is present in the i.i.d. training and test data, the benefit of EU- over AU- and PU-measures increases as the ambiguity modeling in the i.i.d. setting is separated from the EU-measure. This connection can also be observed on GTA5/CS, where the captured AU induced by spatially widespread ambiguities translates to a higher EU-measure performance compared to LIDC.

**General Insights.** Although both, AU- and EU-measures, mostly do behave as expected, the extent of achieved separation depends on the data set properties such as the presence of ambiguities in i.i.d and/or OoD cases. As theoretically motivated in Appendix E, we observe that TTA is in fact most suited for modeling EU, resolving a controversial debate in current literature. We base this on the behavior of our derived EU-measure for TTA being very similar to ensembles and TTD, often even outperforming TTD. The comparable performance to ensembles renders TTA often a cheap alternative for estimating EU. For SSNs, our proposed EU- and AU-measures perform as intended on the toy dataset and GTA5/CS. Whereas on LIDC, the ambiguity, which is mostly present in the border regions, seems to be captured by EU-measures.

## 4.5 RESULTS OF THE EVALUATION ON DOWNSTREAM TASKS

In this section, we address five fundamental questions essential for practitioners when selecting an uncertainty method. The first three concern the components of the uncertainty method: *What is the best (1) uncertainty type, (2) prediction model, and (3) aggregation?* The latter two focus on the robustness of trends: *How robust are settings (4) across datasets and (5) distribution shifts?*

Detailed results can be found in Appendix H. To facilitate parsing the details and to systematically answer the questions from above across downstream tasks, we isolate the performance of each uncertainty type, prediction model, and aggregation method while averaging over the remaining components. Finally, we visualize the performance of the analyzed component with respect to the mean performance on the downstream task for each dataset. The results are shown in Figure 3. We exclude uncertainty types that are not suitable for specific downstream tasks during the averaging process for prediction models and aggregation methods, indicated with crosses in the color of the specific uncertainty measure. Furthermore, we report the standard deviation across the averaged dimensions. *It's essential to recognize that these standard deviations are expected to be relatively high and this indicates a meaningful influence of individual components (C1-C3) on the final performance.*

**OoD-D.** As expected from theory, EU-measures consistently achieve an AUROC above average, mostly outperforming PU-measures as well. Among the prediction models, ensembles are the only model that consistently performs above average across datasets, followed by TTA, which quite ro-

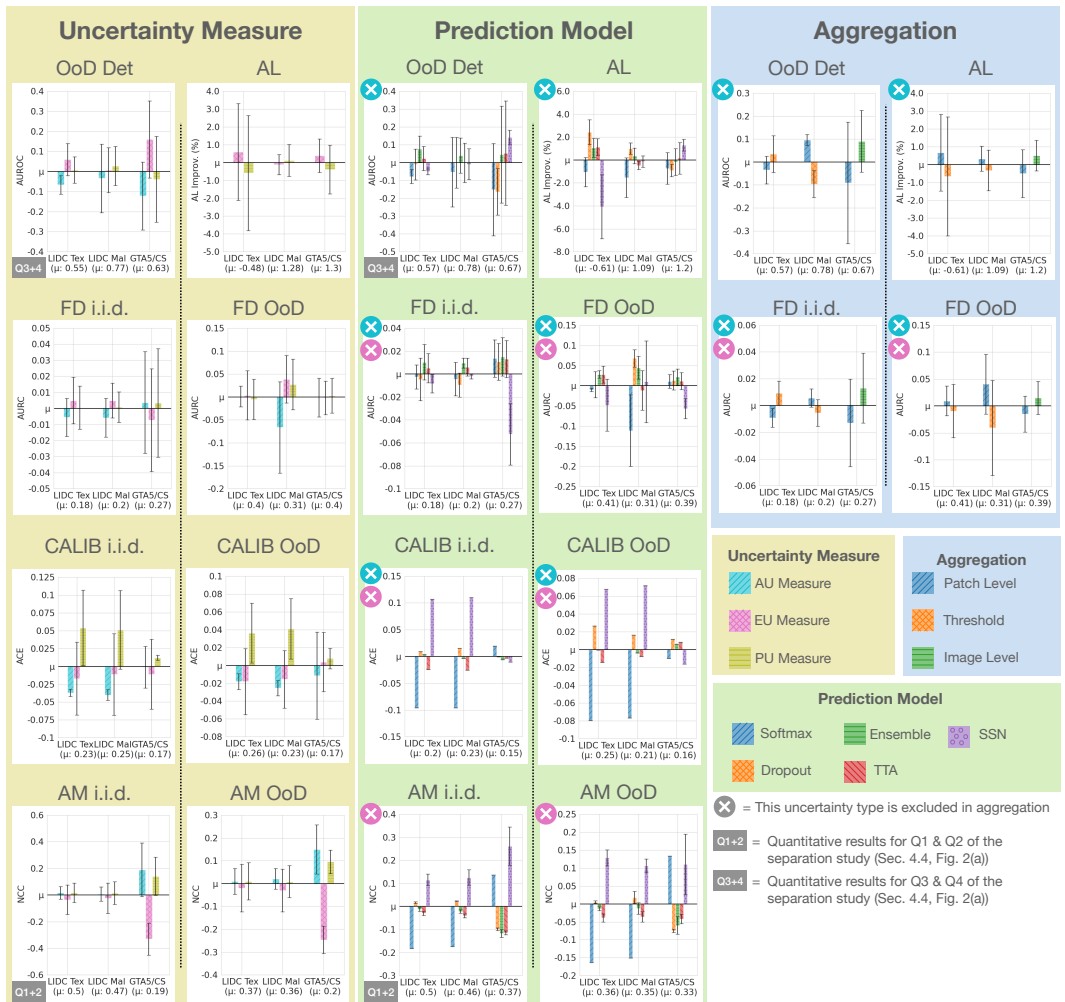

Figure 3: Aggregated results showing deviations from the mean performance to assess general trends for each component (C1-C3) across settings for each dataset. **Due to the averaging across different components (not seeds), high standard deviations are expected.** Uncertainty measures not suited for a specific downstream task are excluded in the average, indicated with crosses in the color of the specific uncertainty measure. Detailed results are shown in Appendix H. Metrics: OoD (AUROC), AL (% improvement over random), FD (AURC), CALIB (ACE), AM (NCC).

bustly performs on or above average. For the GTA5/CS dataset, SSNs outperform other models, as they are able to capture the spatially widespread label ambiguities explicitly and thus isolate the EU. Choosing the optimal aggregation method for this task seems heavily dependent on dataset properties but at the same time very crucial for the performance. This can be seen in the case of LIDC MAL, where differences between aggregation methods exceed standard deviations, emphasizing the substantial influence, regardless of modifications to other components of the uncertainty method.

**Active Learning.** Falling in line with the theoretical perspective, EU-measures generally outperform PU-measures, except on the LIDC MAL task. For prediction models, no model consistently performs above average across all datasets. TTD exhibits strong performance on the LIDC datasets, while SSNs excel on the GTA5/CS dataset. Ensembles are consistently above or close to average performance. The choice of aggregation method exhibits dataset-dependent variability. On the LIDC datasets, patch-level aggregation outperforms threshold aggregation, whereas for the GTA5/CS dataset, image-level aggregation yields the best results. Overall, surpassing the "random" AL baseline appears challenging with marginal improvements on LIDC MAL and GTA5/CS, this observation is in line with recent studies (Mittal et al. (2019; 2023); Lüth et al. (2023)).

**Failure Detection.** Following the theory, PU-measures consistently perform above average on i.i.d data, as failures can arise due to both AU and EU. However, EU proves superior on the LIDC datasets, while AU excels on the GTA5/CS dataset. This trend may be attributed to the larger areas of induced AU in the GTA5/CS dataset, which pose a challenging failure source and increase the importance of AU modeling. Concerning prediction models, ensembles are almost consistently outperforming others, closely followed by TTA. The choice of aggregation method yields mixed results on LIDC TEX, similar to other downstream tasks. Specifically, on i.i.d. data, threshold aggregation is more effective, while patch-level aggregation is better on OoD data. On the other datasets, the trends from the other downstream tasks are confirmed.

**Calibration.** As pixels can be misclassified due to both AU and EU, it is in line with theoretical expectations that PU-measures are consistently the best choice across datasets. For prediction models, TTD performs at least nearly on average or above average compared to other models. Additionally, SSNs exhibit strong performance, particularly on the LIDC datasets. This performance trend remains largely consistent between i.i.d. and OoD data.

**Ambiguity Modeling.** In line with theoretical expectations, AU consistently emerges as the most effective uncertainty type across all datasets. Expectably, this trend is most visible for the large spatial ambiguities induced in the GTA5/CS dataset. When assessing prediction models, SSNs outperform other models, both in i.i.d. and OoD scenarios across all datasets. Notably, SSNs are the only compared model that is specifically designed for AM. One somewhat unexpected observation is the strong NCC performance of the Deterministic model on the GTA5/CS dataset.

**Consistency across datasets and shifts.** Assessing the consistency of best-performing uncertainty methods across datasets, we observe that besides expected results, like EU excelling in OoD detection or SSNs excelling in AM, trends often differ between datasets. This can be especially seen for choosing an appropriate aggregation. Given the low cost of evaluating this post-hoc component, we recommend benchmarking different aggregations. Between the i.i.d. and OoD data, the observed patterns seem more stable, while, as expected, the performance on the OoD data is generally lower.

## 5    CONCLUSION AND TAKE-AWAYS

**General insights & Recommendations.** Our empirical study generates the following insights based on the systematic implementation of requirements R1-R3:

*R1)* When testing the feasibility of separating uncertainty in AU and EU (Sec. 4.4) we found that it works in toy settings but does not necessarily translate to real-world data. In examining the actual benefits of separation (Sec. 4.5) we discovered that these are heavily dependent on the downstream task and the dataset properties. Therefore, neither the feasibility nor the benefit of separation should be taken for granted when presenting a new uncertainty method; instead, convincing evidence should be required for both. Our study shows that such rigorous testing resolves prior contradictions in the literature, e.g. by disproving the assumptions made in Ayhan & Berens (2018) and Wang et al. (2019), revealing that TTA is in fact most suited for modeling EU rather than AU.

*R2)* The explicit validation of individual components C0-C3 shows that in practice it is essential to select optimal components individually based on the dataset properties. One prominent insight is the importance of the aggregation strategy (C3) which can be subject to unwanted correlations and is often oversimplified or neglected in previous work. We show that the choice of C3 is further interdependent on the choices of C1 and C2 and only a joint consideration of all components allows finding the best method configuration for a given task.

*R3)* Our study enables practitioners to make informed choices for all relevant components on their specific task. It also identifies red flags such as the fact that SSNs, while excelling in AM, fall short in FD. Further, the study identifies ensembles as the generally most robust method across downstream tasks, while TTA often represents an adequate and light-weight alternative.

**Impact.** Practitioners can use ValUES to make informed design decisions for uncertainty methods on their problems and methodological developments can be rigorously validated using ValUES, fostering a systematic knowledge base in the field.

ACKNOWLEDGEMENTS

This work was funded by Helmholtz Imaging (HI), a platform of the Helmholtz Incubator on Information and Data Science. This work is supported by the Helmholtz Association Initiative and Networking Fund under the Helmholtz AI platform grant (ALEGRA (ZT-I-PF-5-121)).

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

# A  Downstream Tasks & Metrics

## A.1  Segmentation Performance Assessment

**Dice**  To measure the segmentation performance of the segmentation backbone and prediction models, we used the Dice score which is defined as:

$$\text{Dice}(\hat{y}, y^*) = \frac{2|y^* \cap \hat{y}|}{|y^*| + |\hat{y}|} = \frac{2\text{TP}}{2\text{TP} + \text{FP} + \text{FN}} \tag{2}$$

As we have multiple segmentation predictions $\hat{y}$ for most prediction models and multiple reference segmentations $y^*$, we decided to take the average Dice between each of the $N$ reference segmentations and the mean prediction $\bar{y}$:

$$\text{Dice} = \frac{1}{N} \sum_{i=1}^{N} \text{Dice}(\bar{y}, y_i^*) \tag{3}$$

## A.2  Out of Distribution Detection

In our evaluation, we concentrate on image-level OoD-D to facilitate human assessment, as humans typically evaluate complete images rather than individual pixels. Notably, if any part of an image is identified as OoD, it has the potential to impact all predictions, rendering them unreliable.

**Area Under the Receiver Operating Characteristics Curve (AUROC)**  We calculate the Area Under the Receiver Operating Characteristics Curve (AUROC) to determine a method's capability of detecting OoD cases. Thus, we assign each image a label with 1 equal to an image being OoD and 0 equal to an image being i.i.d. We then use the sklearn library for determining the ROC curve[1] with the ground truth input (0 or 1) and the uncertainty scores as target values. We then determine the AUC also using sklearn[2].

## A.3  Failure Detection

In our evaluation, we concentrate on image-level FD to facilitate human assessment, as humans typically evaluate complete images rather than individual pixels. To this end, we make use of our performance assessment metric on image-level (Dice) to define a continuous failure label for our utilized FD metrics.

Our motivation behind this approach is that FD based on the Dice is more informative with regard to the performance of the model than on the pixel level, as deciding whether a single image should be assessed by a human in place of an automated decision-making process requires to have an assessment of the quality of the segmentation for an entire image than for single pixels.

We compute our FD metrics twice: first using the i.i.d. test data and then the OoD test data. This approach enables us to assess how effectively failures are detected within the i.i.d. data and, subsequently, how well the uncertainty method detects failures when exposed to OoD data.

**Area under the Risk-Coverage-Curve (AURC)**  The Area under the Risk-Coverage-Curve (AURC) is a metric used in selective classification. The goal hereby is to successfully identify failures by having a low *risk*, i.e. a good classifier performance but also achieve high *coverage*, i.e. select as few cases as possible for manual correction. For calculating the Area under the Risk-Coverage-Curve, we use the implementation following Jaeger et al. (2023). To adapt it for a semantic segmentation predictor $f$ and evaluation dataset $D = \{(x_i, y_i)\}_{i=1}^{N}$, we define the *confidence scoring function* (CSF) $g(x_i)$ as the negative uncertainty score. Furthermore, we choose the inverted

---

[1]https://scikit-learn.org/stable/modules/generated/sklearn.metrics.roc_curve.html
[2]https://scikit-learn.org/stable/modules/generated/sklearn.metrics.auc.html

Dice score as the risk $l$ associated with a prediction:

$$l(x, y, f) = 1 - \text{Dice}(f(x), y) \tag{4}$$

The risk-coverage curve is obtained by introducing a confidence threshold $\tau$, which leads to the selective risk

$$\text{Risk}(\tau|f, g, D) = \frac{\sum_{i=1}^{N} l(x_i, y_i, f) \cdot \mathbb{I}(g(x_i) \geq \tau)}{\sum_{i=1}^{N} \mathbb{I}(g(x_i) \geq \tau)} \tag{5}$$

and coverage, defined in Jaeger et al. (2023) as the ratio of cases remaining after selection:

$$\text{Coverage}(\tau|g, D) = \frac{\sum_{i=1}^{N} \mathbb{I}(g(x_i) \geq \tau)}{N} \tag{6}$$

The AURC based on a threshold list $\{\tau\}_{t=1}^{T}$ with $T$ values of a CSF that are sorted ascending can then be calculated as Jaeger et al. (2023):

$$\text{AURC}(f, g, D) = \sum_{t=1}^{T} (\text{Coverage}(\tau_t) - \text{Coverage}(\tau_{(t-1)})) \cdot (\text{Risk}(\tau_t) + \text{Risk}(\tau_{t-1}))/2 \tag{7}$$

where we omitted the conditioning on $f, g, D$ on the RHS for clarity.

**Excess-AURC (E-AURC)**   Further, as also analyzed in Jaeger et al. (2023) and originally proposed in Geifman et al. (2019), we use the *excess AURC* (E-AURC) as an evaluation metric that is independent of the segmentation model's performance:

$$\text{E-AURC} = \text{AURC}(f, g, D) - \text{AURC}(f, g^*, D) \tag{8}$$

where the second term corresponds to the optimal AURC. The optimal CSF $g^*$ can be formally obtained, for example, by using an oracle CSF that returns a confidence equal to the negative risk of a particular prediction, $g^*(x) = -l(x, y, f)$. Practically, it ranks the predictions perfectly by their risk (in our case ascending Dice score). Although we are aware of the fact that evaluating a CSF without considering the performance of the model itself harms the meaningful comparison of uncertainty methods (see Jaeger et al. (2023)), we use this as an additional debugging metric which is feasible in our case as the there are no significant outliers in terms of segmentation performance as seen in the Dice score of Table 5.

### A.4   ACTIVE LEARNING

In our evaluation, we concentrate on AL performing queries on image-level to facilitate human assessment, as humans typically evaluate complete images rather than individual pixels. The general concept here is that we have a model that is already performing well on an i.i.d. dataset with saturated performance for a specific task which should be adapted to a shifted (OoD) dataset with the same task. Therefore we only measure the performance increase on the OoD test set.

**Active Learning Improvement (AL improvement)**   To assess the AL improvement of the uncertainty methods, we measure the relative performance change between two cycles $t_1$ and $t_2$ on the OoD test set:

$$C = \frac{\text{Dice}_{t_2} - \text{Dice}_{t_1}}{\text{Dice}_{t_1}} \tag{9}$$

As we do not want to consider effects of random querying in our evaluation, we subtract the performance change that is reached with random querying from the performance change of the uncertainty method, leading to the following final performance change:

$$C_{\text{final}} = C_{\text{method}} - C_{\text{random}} \tag{10}$$

### A.5   CALIBRATION

Our evaluation of the CALIB follows standard protocol is performed with pixel-level ground truth and aggregated to single images requiring therefore no aggregation.

We compute our CALIB metrics twice: first using the i.i.d. test data and then the OoD test data. This approach enables us to assess how well the uncertainty measure is calibrated on i.i.d. data and, subsequently, how well the uncertainty measure is calibrated when exposed to inputs from OoD data.

**Average Calibration Error (ACE)** The Average Calibration Error (ACE) is introduced in Neumann et al. (2018) and used for segmentation in Jungo et al. (2020). In contrast to the Expected Calibration Error (ECE), which is used e.g. in Gustafsson et al. (2020); Jungo et al. (2020), every bin in the calibration histogram is weighted equally, leading to the following formulation:

$$\text{ACE} = \frac{1}{M} \sum_{m}^{M} |c_m - \text{Acc}_m| \tag{11}$$

Here, $M$ is the number of non-empty bins, $c_m$ is the average confidence in bin $m$, and $\text{Acc}_m$ the respective average accuracy. We apply Platt scaling in order to get confidence scores between $0$ and $1$. We chose this metric in comparison to the ECE as it avoids overweighting the background pixels which are predominant in our case.

### A.6 AMBIGUITY MODELING

Our evaluation of AM is separated into two main parts: first, whether an uncertainty measure can successfully indicate AU in the correct regions, and second, whether a prediction model is able to produce multiple realistic predictions.

The evaluation is performed using pixel-level ground truth based on single images requiring therefore no aggregation.

We compute our AM metrics twice: first using only the i.i.d. test data and on the OoD test data. This approach enables us to assess how good the uncertainty measures model AU on i.i.d. data and, subsequently, how good the uncertainty measures model AU on the OoD data.

**Normalized Cross-Correlation (NCC)** We calculate the normalized cross-correlation (NCC) following Hu et al. (2019):

$$\frac{1}{n_p \sigma_a \sigma_b} \sum_{i=1}^{n_p} (a_i - \mu_a) \cdot (b_i - \mu_b) \tag{12}$$

Here, $a$ is the reference uncertainty map, $b$ is the predicted uncertainty map, $n_p$ is the total number of pixels in the uncertainty maps, and $\mu$ and $\sigma$ are mean and standard deviation of the uncertainty maps. The reference uncertainty map is calculated with the pixel variance of a pixel $y_i$ for $N$ different segmentation raters $\{y_i^1, ..., y_i^N\}$:

$$\mathbb{V}_{p(D)}[y_i] = \frac{1}{N} \sum_{j=1}^{N} (y_i^j - \bar{y}_i)^2 \tag{13}$$

where $\bar{y}_i$ is the mean over the segmentation raters $\bar{y}_i = \frac{1}{N} \sum_{j=1}^{N} y_i^j$.

**Generalized Energy Distance (GED)** To better assess the capability of the uncertainty methods to model multiple raters, we use the generalized energy distance (GED), which has been used in various other works focusing on AM (Kohl et al. (2018); Monteiro et al. (2020); Hu et al. (2019)):

$$D_{\text{GED}}^2(p, \hat{p}) = 2\mathbb{E}_{y \sim p, \hat{y} \sim \hat{p}}[d(y, \hat{y})] - \mathbb{E}_{y, y' \sim p}[d(y, y')] - \mathbb{E}_{\hat{y}, \hat{y}' \sim \hat{p}}[d(\hat{y}, \hat{y}')] \tag{14}$$

Here, $d(y, y')$ is the distance between two reference segmentations, and $d(\hat{y}, \hat{y}')$ is the distance between two predicted segmentation variants. $p$ and $\hat{p}$ are the respective reference and predicted distributions for the segmentations masks. The distance has to satisfy that it increases for more dissimilar masks and further $d(x, y) = 0$ for $x = y$. As we use the Dice as our main evaluation metric, we chose to use $d(x, y) = 1 - \text{Dice}(x, y)$ as distance.

# B  DATASETS

## B.1  TOY DATASET SETUP

### B.1.1  DATASET SCENARIOS

As mentioned in Sec. 4.2, we generate three different training and four different testing scenarios for the toy dataset. An overview of the different scenarios, including the number of training and testing cases in each scenario, is shown in Table 1. Each of those settings is targeted at answering a specific question in our separation study, described in Sec. 4.1. In setting 1, we induce AU, thus aiming to answer Q1 and Q2 of the separation study. Setting 2 focuses on EU, and thus aims to answer Q3 and Q4. However, since AU is not induced in setting 2, we hypothesize that the behavior of AU-measures should not be well-predictable, limiting the ability to clearly answer Q4. Therefore, we design setting 3, and provide testing scenarios (a) and (b) where AU is induced during training and (b) where AU is also present in the i.i.d testing data. These aim at understanding the behavior of our uncertainty measures to detect EU with varying degrees of AU present.

Table 1: Number of training and testing cases for the toy dataset. For each scenario, the number of training cases and the number of testing cases is specified. Further, the number of cases with ambiguity / blur is specified in brackets and the number of i.i.d and OoD cases in the testset.

| Scenario | Description | # Train (# blur) | # Test | |
|---|---|---|---|---|
| | | | # i.i.d (# blur) | # OoD |
| 1 | Training models on data with induced AU; testing on i.i.d. data also containing AU | 200 (200) | 20 | |
| | | | 20 (20) | 0 |
| 2 | Training models on data without ambiguity; testing on i.i.d. data and shifted data | 200 (0) | 42 | |
| | | | 21 (0) | 21 |
| 3a | Training models on data with and without blur/ambiguity; testing on i.i.d. data and shifted data without blur | 200 (100) | 42 | |
| | | | 21 (0) | 21 |
| 3b | Training models on data with and without blur/ambiguity; testing on i.i.d. data and shifted data without blur and i.i.d data with blur | 200 (100) | 63 | |
| | | | 42 (21) | 21 |

### B.1.2  DATA WITH INDUCED ALEATORIC UNCERTAINTY

Figure 4 shows the data scenario that is created with induced aleatoric uncertainty. The input (Figure 4a) shows a sphere that has Gaussian blur to the outside. Due to the blur to the outside, the exact border of the sphere is ambiguous. This ambiguity is modeled by three different reference raters (Figure 4b - Figure 4d). Thereby the segmentation of rater 1 (Figure 4b), that segments the smallest sphere, is 10% the size of the segmentation of rater 3 (Figure 4d). Rater 2 (Figure 4c) lies exactly between those two raters, so the size of its segmentation is 55% the size of rater 3.

The test set (Figure 4e) is created in the same manner as the training set. The expected uncertainty is shown in Figure 4f with the respective legend in Figure 4g. No epistemic uncertainty should be present in the data when the model converged after training because the test set is created identically to the training set. Instead, only aleatoric uncertainty should be present. This aleatoric uncertainty is expected to be in the ambiguous area of the border of the sphere.

### B.1.3  DATA WITH INDUCED EPISTEMIC UNCERTAINTY

Figure 5 shows the data scenario that was created with induced epistemic uncertainty. The input object in the training data (Figure 5a) is a sphere, like in the dataset with aleatoric uncertainty.

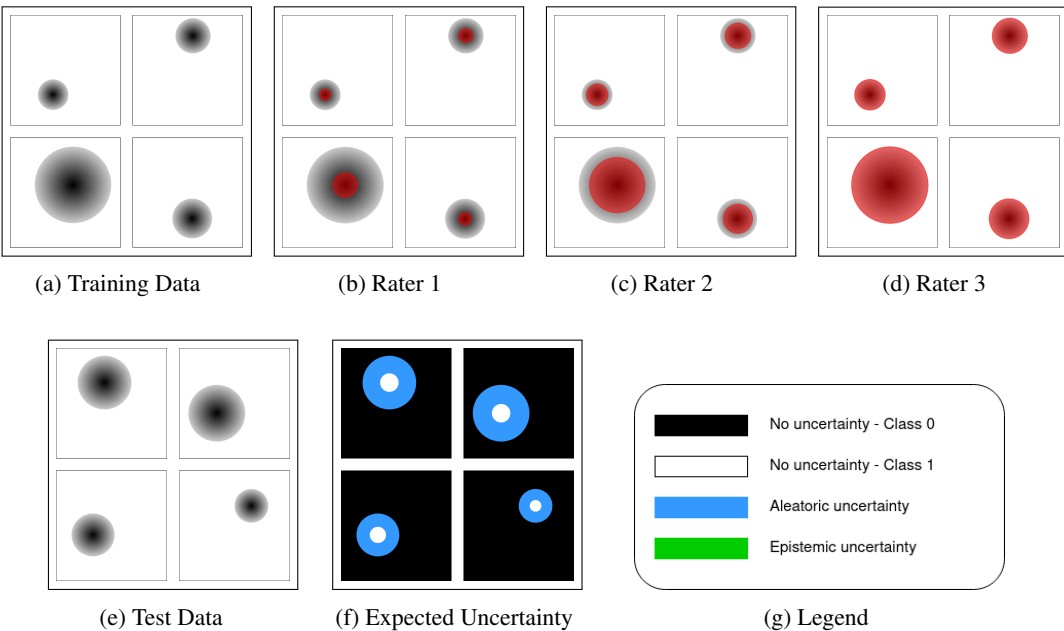

(a) Training Data  (b) Rater 1  (c) Rater 2  (d) Rater 3

(e) Test Data  (f) Expected Uncertainty  (g) Legend

Figure 4: Aleatoric data scenario. (a) shows the input images in the training set, which are ambiguous due to Gaussian blur to the outside. (b) - (d) show three different reference ratings that are generated for the input images. (e) shows test images and (f) the expected uncertainty maps. The uncertainty regions are explained in (g).

However, for the epistemic data scenario, this sphere has no blur to the outside, to define a clear segmentation boundary for the ground truth segmentation (Figure 5b). As the segmentation problem would be too simple if the background was just black, random noise was added to the background for this case. The test set for this dataset is shown in Figure 5c. It consists of objects of different shapes and colors that were not present in the training data. Some of the objects are still spheres but with varying gray values. Furthermore, there are cubes in the test set and spheres that are partially outside the image while the spheres in the training set were always fully inside the image.

As all segmentations are unique, no aleatoric uncertainty should be present in the data. However, where exactly to expect epistemic uncertainty is not that clear. In some cases, the network might generalize, depending on which features were mainly learned during the training (Geirhos et al. (2020)). For the given toy example it is unknown which training solution the network learned. If it learned to recognize the shape of the object, new shapes should yield a higher uncertainty in the prediction, as shown in Figure 5d. On the other hand, if the network learned the intensity, the epistemic uncertainty might look more like in Figure 5e. It might also be the case that the network learned another decision rule which might result in a different epistemic uncertainty.

## B.2 LIDC-IDRI DATASET SETUP

### B.2.1 DATASET PREPROCESSING

For preprocessing the dataset, we use the pylidc library (Hancock & Magnan (2016)). With this library, all nodules with size $\geq 3\,\mathrm{mm}$ can be queried and clustered, such that each nodule gets assigned up to four raters. We ignore cases that are so close together that they cannot be grouped to one nodule automatically. Further, we calculate a consensus mask which is the union of all raters and ignore cases that have a consensus mask larger than $64$ voxels in one direction. We crop patches of size $64 \times 64 \times 64$ with the nodule in the center and all images are resampled to a resolution of $1 \times 1 \times 1\,\mathrm{mm}$. Also, we only consider nodules in our following analysis that have four reference segmentation masks, which are overall $901$ nodules.

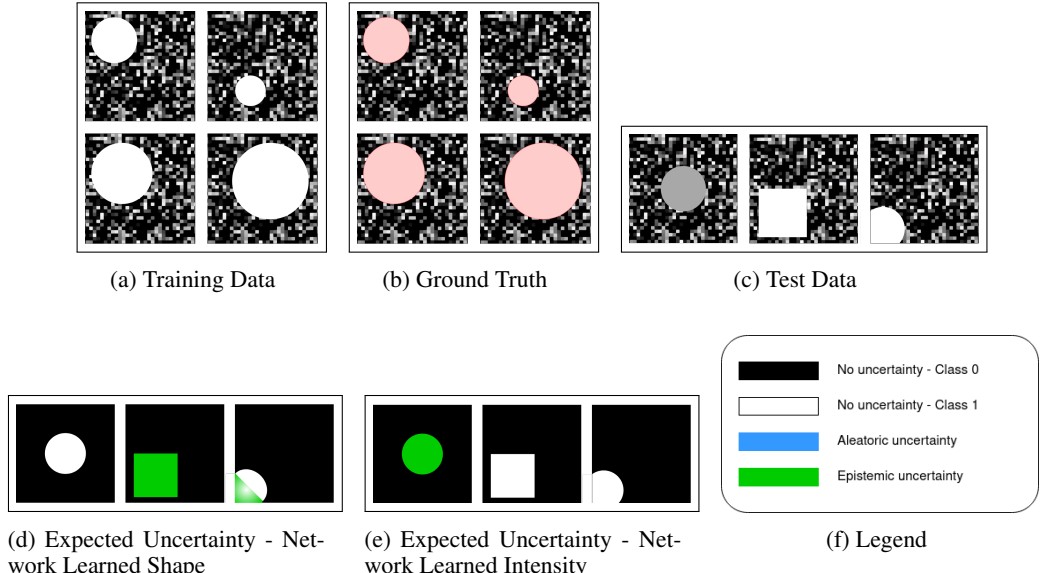

(a) Training Data      (b) Ground Truth      (c) Test Data

(d) Expected Uncertainty - Network Learned Shape     (e) Expected Uncertainty - Network Learned Intensity     (f) Legend

Figure 5: Epistemic data scenario. (a) shows the input images in the training set. (b) shows the ground truth segmentation. (c) shows test images that differ in various aspects from the training data. (d) and (e) show possible uncertainty maps, depending on what the network learned. The uncertainty regions are explained in (f).

### B.2.2 METADATA DISTRIBUTION SHIFT ANALYSIS

Overall, the dataset contains nine different features described in the metadata: *subtlety*, *internal structure*, *calcification*, *sphericity*, *margin*, *lobulation*, *spiculation*, *texture* and *malignancy*. All of these features contain 4-6 different possible categories. Each segmentation rater assigned one of the categories to the metadata features. For inducing distribution shifts, we binarize each metadata feature into two classes (i.i.d. and OoD) instead of the original categories. To not have too much entanglement with aleatoric uncertainty in the distribution shift analysis, we leave out the features *subtlety* and *margin* because if a nodule is subtle, it might be likely that it is not labeled by all raters and if the margin is not sharp, there might be a high variability at the border of the nodule. Further, we do not consider the feature *internal structure*, as it has only one OoD case which makes it unsuitable for a comparison between i.i.d. and OoD cases.

Next, we construct a train/test split to analyze the performance difference of a deterministic U-Net model on the i.i.d. test set and the OoD test set. The way this split is constructed is shown in Figure 6. We first remove all nodules that do not have a majority vote for being i.i.d. or OoD, i.e. when two raters voted for the nodule being i.i.d. and two voted for the nodule being OoD. Next, all patients are identified that have at least one OoD nodule. The OoD nodules of these patients are added to the OoD test set and the i.i.d. nodules of these patients are added to the i.i.d. test set. From the remaining patients that only have i.i.d. nodules, most of the nodules are taken in the i.i.d. training set and some nodules are added to the i.i.d. test set, such that the overall ratio of i.i.d. nodules in the training set and the i.i.d. test set is $80\%/20\%$. The split which cases to include in the training set and the i.i.d. test set is decided by the patient identifier. With the described approach for creating the splits, it is ensured that no patient has nodules in the training- and the test set at the same time.

To measure the performance drop between the i.i.d. and the OoD test set, 5 folds are trained for every metadata split with varying seeds between the folds. The mean Dice between the prediction and one random rater and the standard deviation are calculated on the i.i.d. and the OoD test set to measure the performance. The results are shown in Table 2.

After determining the performance drop on each feature, the two features with the highest performance drops are selected to examine in further experiments. These are the texture shift and the malignancy shift. It can be seen from the results that there is a substantial drop between the i.i.d. and OoD performance which confirms the approach for inducing epistemic uncertainty.

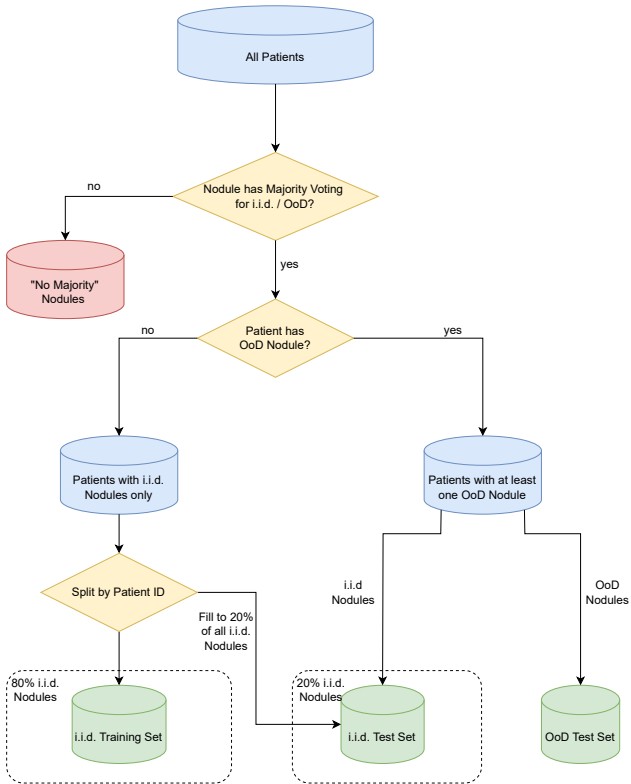

Figure 6: Splits for the LIDC-IDRI shift analysis. Only nodules are considered that have a majority vote for either being i.i.d. or OoD. Furthermore, the splits are created considering the patient ID, so that no nodules of the same patient are in the training set and the test set at the same time. In the end, an i.i.d. training set, an i.i.d. test set, and an OoD test set are created to analyze the shifts between the features.

### B.2.3   SETUP FOR EVALUATION ON DOWNSTREAM TASKS

To evaluate the performance on the various downstream tasks, the lung nodules are divided into three sets: i.i.d. training set, i.i.d. and OoD test set, and i.i.d. and OoD unlabeled pool. The size of these sets is shown in Table 3. Initially, we only train the model on the i.i.d. training set and assume that its performance on i.i.d. data reaches saturation. After this training, we evaluate the

Table 2: Results for the LIDC-IDRI shift analysis. For each feature, the Dice score on the i.i.d. test set, the Dice score on the OoD test set and the performance drop between i.i.d. and OoD test set are shown. Mean and standard deviation are reported for training with 5 folds, each with a different seed.

| Feature | i.i.d. / OoD | Dice i.i.d. | Dice OoD | Performance Drop (%) |
|---|---|---|---|---|
| Calcification | Absent / Present | $0.804 \pm 0.0022$ | $0.7669 \pm 0.0133$ | $4.6112 \pm 1.7699$ |
| Sphericity | Round / Linear | $0.7934 \pm 0.0042$ | $0.7474 \pm 0.0106$ | $5.7905 \pm 1.191$ |
| Lobulation | No Lobulation / Lobulation | $0.7887 \pm 0.0042$ | $0.7649 \pm 0.0046$ | $3.0163 \pm 0.6264$ |
| Spiculation | No Spiculation / Spiculation | $0.7958 \pm 0.0022$ | $0.7458 \pm 0.0068$ | $6.2865 \pm 0.9447$ |
| Texture | Solid & Part Solid / Non-solid | $0.81 \pm 0.0012$ | $0.6081 \pm 0.0124$ | $24.9244 \pm 1.431$ |
| Malignancy | Non-malignant / Malignant | $0.7789 \pm 0.0051$ | $0.6677 \pm 0.0645$ | $14.3093 \pm 7.9522$ |

performance of the uncertainty methods on FD, CALIB, and AM. Then, we select samples from the unlabeled pool based on uncertainty rankings. Based on this uncertainty ranking on the unlabeled pool, we determine the performance of a method for detecting OoD samples and add the highest 50% of uncertain samples to the training pool, aiming for improved performance on the OoD test set. With this modified training set, we train another iteration and afterward again measure the test set performance.

Table 3: Size of the different sets in the LIDC dataset for the evaluation on the various downstream tasks.

| Split | Train | Val | Test | | Unlabeled Pool | |
|---|---|---|---|---|---|---|
| | | | i.i.d | OoD | i.i.d | OoD |
| Texture | 513 | 129 | 167 | 20 | 42 | 20 |
| Malignancy | 200 | 51 | 105 | 93 | 184 | 92 |

### B.3 GTA5/CITYSCAPES DATASET SETUP

As a further dataset, we use a combination of the GTA5 dataset (Richter et al. (2016)) and the Cityscapes dataset (Cordts et al. (2016)). As mentioned in Sec. 4.2, both datasets contain the same classes, and thus we use the GTA5 dataset as i.i.d. data and the Cityscapes dataset as OoD data.

From the Cityscapes dataset, we use the training set as an unlabeled pool for the AL downstream tasks and the validation set as test set. The scheme for splitting the datasets into training, test set, and unlabeled pool is thereby shown in Figure 7, with the concrete number of images per split. Concretely, we randomly select the same amount of images from the GTA5 dataset to be in the unlabeled pool and further create a 75/25 training/testing split for the GTA5 dataset.

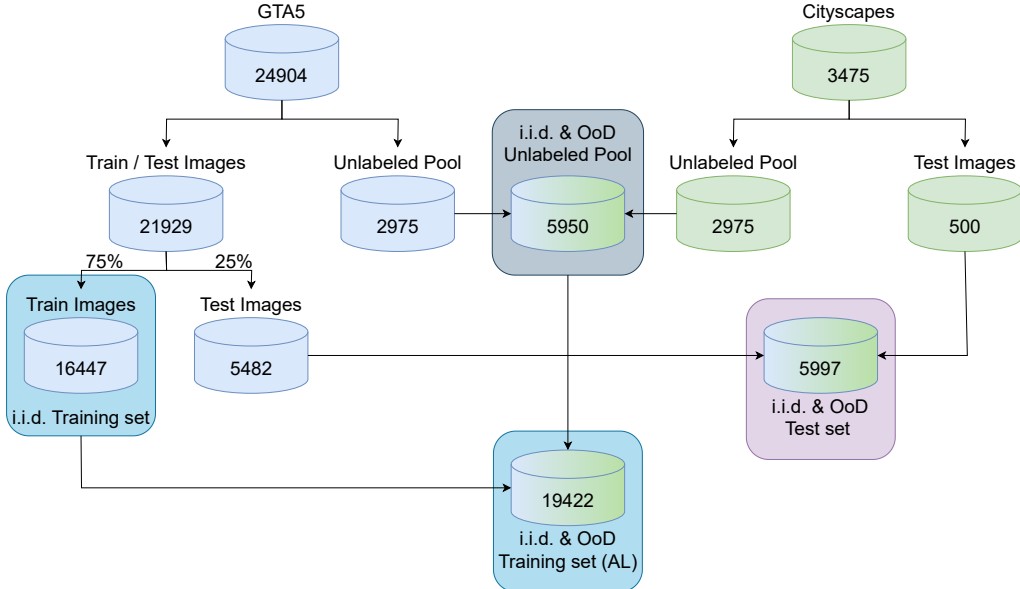

Figure 7: Splits for the GTA5/CS dataset. From the Cityscapes dataset, the training set is used as unlabeled pool and the validation set is used as test set.

The Cityscapes dataset contains up to 30 classes, but only 19 of them are used for validation. As only those 19 classes are contained in the GTA5 dataset, we restrict our analysis to these classes. Further, as mentioned in Sec. 4.2, we perform random class switches for the classes "sidewalk", "person", "car", "vegetation" and "road" with a probability of $\frac{1}{3}$ from <class> to <class 2>. This approach is also applied by Kohl et al. (2018).

All images are first cropped to a size of $1024 \times 1912$ and then rescaled to $25\%$ of the size, resulting in an image size of $256 \times 478$.

## C  MODEL IMPLEMENTATION DETAILS

In this section, we describe the implementation details of the model backbones and prediction models. It is important to note that we did not do extensive hyperparameter searches for all the hyperparameters that we state here but rather used standard values if they worked reasonably and if not, we performed sweeps over the hyperparameters to find appropriate settings. We are aware that especially in the implementation of the prediction models, many hyperparameters are involved, and exploring more hyperparameter settings might be an interesting direction for future experiments. All models are trained for 150 epochs.

### C.1  SEGMENTATION BACKBONES

**U-Net**  For the toy dataset and the LIDC datasets, we use a 3D U-Net architecture as segmentation backbone. We thereby use an initial filter size of 8 for the toy dataset and 16 for the LIDC datasets and four encoder and four decoder blocks. As loss function, we use a combination of the Dice loss and the cross-entropy loss except for the SSNs as prediction model. For the SSNs, we use the loss function as specified in Monteiro et al. (2020). The Adam optimizer is used with a learning rate of $3e-4$ and a weight decay of $1e-5$. The batch size is set to 8. As augmentations, we apply random flipping and Gaussian noise.

**HRNet**  For the GTA5/CS dataset, we use the HRNet as segmentation backbone, pretrained on ImageNet. As loss function, we use the cross-entropy loss, again, except for the SSNs. For all prediction models except SSNs, SGD is used as optimizer with a learning rate of $0.01$, weight decay of $5e-4$, and momentum of $0.9$. For the SSNs, RMSprop is used as optimizer with a learning rate of $1e-4$, weight decay of $5e-4$, and momentum of $0.6$. The batch size is set to 6. As augmentations, we use random horizontal flipping, rotations, random scaling, random cropping, and Gaussian noise.

### C.2  PREDICTION MODELS

**Test-time dropout (TTD)**  For the U-Net, we add dropout after each convolutional block with a probability of $p = 0.5$. For the HRNet, we add dropout at the end of each branch, following Nash et al. (2022). Again, the probability is set to $p = 0.5$. During inference, we perform 10 MC-Dropout forward passes for each input.

**Ensemble**  For the ensemble models, we do not change anything about the models and training schemes themselves but train 5 models with different seeds. During inference, we pass each input image through all 5 models.

**Test-time data augmentations (TTA)**  For the TTA models, we apply the same augmentations as used in training for the 3D U-Net. Thereby, we apply all possible combinations of flipping and Gaussian noise, which result in 16 forward passes per input image (8 possible flipping directions, each with and without noise). For the HRNet, we also apply all possible combinations of random horizontal flipping and Gaussian noise, resulting in 4 forward passes per input image (2 flipping possibilities, each with and without noise).

**Stochastic Segmentation Networks (SSNs)**  For the stochastic segmentation networks, we do 10 forward passes per input image. For the toy dataset and the LIDC datasets, we use a rank of 5 and for the GTA5/CS dataset, we use a rank of 10. As the training behaved more stable when pretraining the mean first, we perform 5 pretraining epochs where we only train the mean before we also train the covariance matrix.

# D   UNCERTAINTY MEASURES FOR PROBABILISTIC VARIABILITY VARIABLE PREDICTION MODELS

For a probabilistic prediction model $p(Y|x) = \mathbb{E}_{z \sim p(z)}[p(Y|x, z)]$ which predicts the class variable $Y$ given a sample $x$ with an additional variable $Z$ following $p(z)$ which is supposed to capture the variability of the raters/labels (variability variable), we hypothesize that AU und EU can be estimated in a similar fashion as it is done for Bayesian models following

$$\underbrace{H(Y|x)}_{\text{PU}} = \underbrace{\text{MI}(Y, Z)}_{\text{AU (for i.i.d. } x)} + \underbrace{\mathbb{E}_{z \sim Z}[H(Y|z, x)]}_{\text{EU}}. \tag{15}$$

Examples of these methods are the SSNs (Monteiro et al. (2020)), the probabilistic U-Net (Kohl et al. (2018)) or PHiSeg (Baumgartner et al. (2019)) where the prediction model is trained explicitly to learn the variability of the raters.
A more detailed motivation is given in the following two paragraphs and a reason for our observed failure mode is described in the third paragraph.

**Aleatoric uncertainty.**   Multiple plausible predictions for a sample due to ambiguity or other factors are commonly attributed as AU (Monteiro et al. (2020); Kendall & Gal (2017)) and therefore leads to the assumption that the variability variable $Z$ essentially captures the learned AU of the prediction model. Therefore the mutual information between the class label $Y$ and the variability variable $Z$ given a sample $x$ describes how much information about the AU could be gained by obtaining the class label $y$.

$$\text{MI}(Y, Z|x) = H(Y|x) - \mathbb{E}_{z \sim Z}[H(Y|x, z)] \tag{16}$$

Knowing the optimal variability variable $Z$ would essentially lead to alleviating the uncertainty. Therefore we hypothesize that this uncertainty measure models AU.

**Epistemic uncertainty.**   Following the notion that there is no reason for a variability variable prediction model ever to be unsure about its prediction on i.i.d. data if it is still dependent on the variability variable $p(Y|x, z)$ [3] Therefore the uncertainty of the classifier $H(Y|x)$ which can not be attributed to the variability variable $Z$ should be novel and previously unseen (by the prediction model). Following this line of reasoning, we hypothesize that the expected entropy of the variability variable models EU.

$$\mathbb{E}_{z \sim Z}[H(Y|x, z)] = H(Y|x) - \text{MI}(Y, Z|x) \tag{17}$$

**Failure mode.**   For the SSNs, we observe in our experiments that the model while still dependent on the variability variable is often uncertain in border regions between two classes but generally does not extend to large regions of the image [4]. This offers an explanation why for the experiments on the LIDC-IDRI dataset, where for most samples the disagreement between raters is present purely in the border regions of the nodule, $\text{MI}(Y, Z|x)$ has the lowest NCC scores (Q1 + Q2).

# E   UNCERTAINTY MEASURES FOR TEST-TIME AUGMENTATION MODELS

Given a model using a set label preserving data augmentations during inference which are defined on the input space $\mathcal{T}$ and used the form of a random variable $T$ ($\text{support}(T) = \mathcal{T}$) from which samples are drawn from $t \sim T$. The inference using test-time augmentations can can be described as $p(Y|x) = \mathbb{E}_{t \sim T}[p(Y|t, x)] = \mathbb{E}_{t \sim T}[p(Y|t(x))]$. During training the model is optimized on the training set $\mathcal{D}$ with a training objective (usually the cross-entropy loss (CE-Loss)) to be invariant against augmentations in $\mathcal{D}$. Given an optimal model for which the training objective is minimal (e.g. CE-Loss=0), the outputs of the model on the training set $\mathcal{D}$ are fully invariant to all transformations $p(Y|x, t_1) = p(Y|x, t_2) \forall t_1, t_2 \in \mathcal{T}, x \in \mathcal{D}$.

---

[3]This is essentially designed into the training of the variability variable prediction models. E.g. for the SSNs, this is done so by using the logsumexp of the logarithmic loss Monteiro et al. (2020).

[4]We hypothesize that this behavior arises due to $p(z)$ modeling a Gaussian distribution in logit space.

For this model, we hypothesize that AU und EU can be estimated in a similar fashion as it is done for Bayesian models following

$$\underbrace{H(Y|x)}_{\text{PU}} = \underbrace{\text{MI}(Y,T)}_{\text{EU}} + \underbrace{\mathbb{E}_{t \sim T}[H(Y|t,x)]}_{\text{AU (for i.i.d. } x)}. \tag{18}$$

A more detailed motivation is given in the following two paragraphs.

**Aleatoric uncertainty.** As our model is perfectly trained on the training set, the model is able to detect previously seen uncertainty over the augmentations similar to a Bayesian model can do so with each set of parameters Mukhoti et al. (2021). Therefore, the expected entropy over the augmentations should give information about the amount of AU in the prediction of a datapoint.

$$\mathbb{E}_{t \sim T}[H(Y|x,t)] \tag{19}$$

**Epistemic uncertainty.** As our model is invariant to augmentations on the training set it also follows that $\text{MI}(Y,T|x) = 0 \forall x \in \mathcal{D}$ Jensen (1906). If the mutual information between the augmentation variable and predicted label is greater than zero ($\text{MI}(Y,T|\hat{x}) > 0$) for a datapoint $\hat{x} \notin \mathcal{D}$, then this indicates that this datapoint deviates in some form from $\mathcal{D}$. Further, if $\hat{x}$ would be added to the training set and the model retrained, the model would have learned to be invariant against the augmentations for this datapoint. Following this argumentation, this term is therefore reducible by adding previously unseen datapoints. Based on this, we hypothesize that the mutual information between the augmentation variable and the predicted label models EU.

$$\text{MI}(Y,T|x) = H(Y|x) - \mathbb{E}_{t \sim T}[H(Y|x,t)] \tag{20}$$

**Implications.** Based on these derivations it seems that TTA actually allows the model to estimate EU, rather than improving the estimation of AU. This falls in line with the hypothesis made by Hu et al. (2019) and directly opposes the claims of two prominent papers claiming it models AU (Wang et al. (2019); Ayhan & Berens (2018)).

## F  DETAILS ON THE AGGREGATION STRATEGIES

### F.1  ABLATION STUDY: CORRELATION OF IMAGE LEVEL AGGREGATION AND OBJECT SIZE

To confirm the hypothesis about the correlation between the object size and the amount of uncertainty, we generated plots to see the connection between those two variables for the LIDC datasets. One of the generated plots is shown in Figure 8. This plot is for a TTD model on the LIDC TEX dataset. In the top row, the aggregated amount of uncertainty compared to the mean size of the predicted segmentation is shown for the epistemic, the aleatoric, and the predictive uncertainty. In the bottom row, the summed uncertainty is divided by the object size. To confirm that the size of the predicted segmentation corresponds to the ground truth segmentation size, the two variables are plotted on the right-hand side. It can be seen, that a positive correlation between the aggregation sum and the object size is given in the top row, but if the aggregation mean is taken in the bottom row, this correlation is not present. This means that the summed uncertainty only correlates with the size of the objects and does not represent the objects' uncertainty independent of the size.

### F.2  SELECTION OF THRESHOLD FOR THRESHOLD LEVEL AGGREGATION

For the threshold level aggregation, we need to determine a threshold where the pixels that are above this are considered as "uncertain". Intuitively, most uncertainty is likely to be at the border of the object and thus correlates with the object size. Therefore, the threshold is calculated with respect to the object sizes in the validation set in the following way: First, the mean foreground ratio $\alpha$ over all predicted segmentations in the validation set is determined:

$$\alpha = \frac{\#\text{voxels foreground pred}}{\#\text{voxels}} \tag{21}$$

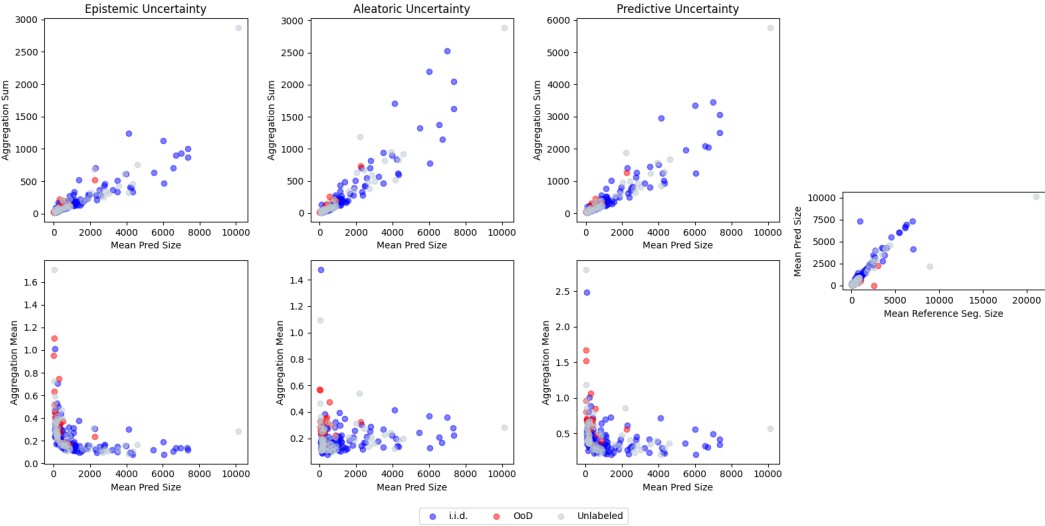

Figure 8: Correlation between object size and uncertainty for image level aggregation. In the top row, all pixels in the uncertainty maps are added up and this aggregation sum is plotted with respect to the mean size of the predicted segmentations. In the bottom row, the aggregation sum is additionally divided by the predicted object size, resulting in the aggregation mean. On the right-hand side, the mean prediction size is plotted with respect to the mean reference segmentation size to see that the size of the predictions roughly corresponds to the reference segmentation sizes of the objects.

With this foreground ratio, the quantile value $q$ is calculated with $q = 1 - \alpha$. This quantile value is applied on the predicted uncertainty maps of the validation set $u_{\text{val}}$, to determine a pixel value of pixels that lie in that quantile $Q$. This pixel value then serves as a threshold for later predicted images:

$$\text{threshold} = Q(q, u_{\text{val}}) \tag{22}$$

With this method, one threshold per uncertainty modeling method is determined.

# G    DETAILED RESULTS OF THE SEPARATION STUDY

## G.1    DETAILED ANALYSIS

Q1 & Q2

*Toy dataset.* In the toy dataset analysis, AU uncertainty measures have generally higher NCC values compared to EU uncertainty measures, indicating a successful separation of AU and the highlighting of relevant areas (Q1) which is also supported by the qualitative analysis with high uncertainty signals in areas with rater disagreement (see Sec. G.3). Meanwhile, the EU-measures perform worse than PU and AU measures indicating that EU-measures do not measure AU. An exception to this finding are SSNs, where NCC scores are higher for EU-measures compared to other prediction models. This discrepancy may be attributed to the presence of AU at the border regions which is not explained by the variability variable.

*LIDC datasets.* On the LIDC datasets, EU-measures performance is similar to that of AU-measures. Therefore the approaches seem to model EU in the areas attributed to AU (Q2).
In fact, for SSNs, the AU-measure even shows a lower NCC than the EU-measure, which could be attributed to the SSNs rating the border regions with high EU, which are the regions of disagreement. The qualitative analysis shows that a slightly better indication of AU by AU-measures becomes apparent when there is meaningful inter-rater variability beyond small border regions (see Figure 10). Interestingly, this effect is only noticeable for i.i.d. nodules. For example, the same nodule that

shows a good indication for AU in the i.i.d. test set on the LIDC TEX dataset (Figure 10) shows a poor indication of AU in the OoD test set on the LIDC MAL dataset (Figure 13).

*GTA5/Cityscapes dataset.* For the GTA5/CS dataset, the NCC scores are generally lower compared to the other datasets. However, for AU-measures, the NCC scores are at least positively correlated, while the EU-measures are mostly even negatively correlated with the AU, showing that they really do not model AU. The only prediction that reaches a high AU with its respective AU-measure are SSNs. This qualitative difference can also be seen in Figure 14: While most prediction models show the highest AU at the borders of the object, SSNs AU-measure highlight the whole ambiguous area.

Q3 & Q4

*Toy dataset.* In Setting 2, where only EU is present in the data, there is no significant difference in AUROC between AU-measures and EU-measures. It could be assumed that in the absence of learning AU in the training data, every uncertainty measure can be interpreted as EU-measure. However, as soon as AU is introduced into the training data in settings 3a and 3b, EU-measures become a better separator between i.i.d and OoD data. In setting 3b, where AU is present in both the training and test data, the separation of EU becomes beneficial. To address Q3 and Q4 on the toy dataset, it can be seen that the AUROC retrieved with EU-measures is almost always better than random, confirming Q3. The answer to Q4 depends on the amount of AU present in the training and test data.

*LIDC datasets.* On the LIDC datasets, it is evident that the separation between AU and EU brings particular benefits for TTD, Ensembles, and TTA. Specifically, on LIDC TEX, EU-measures prove to be a more effective separator between i.i.d. and OoD data. Overall, it can be seen that whenever the separation between PU and EU is advantageous, AU as an OoD-detector performs worse than random. Another hypothesis that arises is that the separation of EU appears to be most beneficial in settings where the OoD-detection performance is not yet saturated, such as in the case of LIDC TEX. To summarize the answer for Q3 and Q4 for the LIDC dataset, the AUROC for EU-measures is always better than random, confirming Q3. However, Q4 can only be partially confirmed in the sense that AU is not a good measure whenever separating EU from PU is beneficial.

*GTA5/Cityscapes dataset.* For the GTA5/CS dataset, most EU-measures significantly outperform the respective AU-measure by means of the AUROC. The only exception is TTD, where the EU-measure even performs worse than the AU-measure. Besides that, the AU-measures are even below random performance for the patch-level aggregation, while for the image-level aggregation, they perform slightly better than random. This indicates in summary with regards to Q3, that EU-measures, except for TTD, capture EU, while for Q4, it can be at least mostly confirmed that AU-measures do not consistently outperform random selection of OoD cases.

### G.2    QUANTITATIVE RESULTS

The detailed quantitative results for the separation study, presented in Sec. 4.4, can be found in Table 4. Table 4a provides insights on answering Q1 and Q2, while Table 4b addresses Q3 and Q4.

Table 4: Quantitative results for the separation study. In order to answer Q1 and Q2 from the separation study, the NCC scores are calculated between the uncertainty maps and the variance of the reference segmentations, shown in Table 4a. To answer Q3 and Q4, the AUROC scores are calculated and reported in Table 4b. Mean results are shown over 3 runs with different seeds for all relevant dataset settings to answer the respective questions. Abbreviations: PM: Prediction model, UM: Uncertainty measure, UT: Modeled uncertainty Type (according to theory), AGG: Aggregation strategy.

| Testset | PM | UM | UT | Toy 1 | LIDC TEX | LIDC MAL | GTA5/CS |
|---|---|---|---|---|---|---|---|
| i.i.d | Determ. | MSR | PU | 0.68 | 0.32 | 0.28 | 0.51 |
| | TTD | PE | PU | 0.80 | 0.51 | 0.48 | 0.27 |
| | | EE | AU | 0.86 | 0.52 | 0.48 | 0.28 |
| | | MI | EU | 0.47 | 0.46 | 0.45 | -0.23 |
| | Ensemble | PE | PU | 0.83 | 0.48 | 0.43 | 0.24 |
| | | EE | AU | 0.84 | 0.49 | 0.44 | 0.27 |
| | | MI | EU | 0.51 | 0.39 | 0.36 | -0.23 |
| | TTA | PE | PU | 0.82 | 0.46 | 0.41 | 0.25 |
| | | EE | AU | 0.82 | 0.48 | 0.42 | 0.26 |
| | | MI | EU | 0.54 | 0.38 | 0.35 | -0.16 |
| | SSN | PE | PU | 0.96 | 0.63 | 0.61 | 0.56 |
| | | MI | AU | 0.96 | 0.59 | 0.55 | 0.70 |
| | | EE | EU | 0.80 | 0.64 | 0.62 | 0.05 |
| OoD | Determ. | MSR | PU | - | 0.20 | 0.20 | 0.47 |
| | TTD | PE | PU | - | 0.37 | 0.36 | 0.26 |
| | | EE | AU | - | 0.37 | 0.39 | 0.26 |
| | | MI | EU | - | 0.33 | 0.31 | -0.13 |
| | Ensemble | PE | PU | - | 0.35 | 0.33 | 0.25 |
| | | EE | AU | - | 0.36 | 0.35 | 0.30 |
| | | MI | EU | - | 0.30 | 0.27 | -0.06 |
| | TTA | PE | PU | - | 0.32 | 0.30 | 0.28 |
| | | EE | AU | - | 0.33 | 0.33 | 0.30 |
| | | MI | EU | - | 0.27 | 0.25 | -0.04 |
| | SSN | PE | PU | - | 0.51 | 0.47 | 0.37 |
| | | MI | AU | - | 0.47 | 0.44 | 0.52 |
| | | EE | EU | - | 0.52 | 0.47 | 0.03 |

(a) NCC scores

| PM | UM | UT | AGG | Toy 2 | Toy 3a | Toy 3b | LIDC TEX | LIDC MAL | GTA5/CS |
|---|---|---|---|---|---|---|---|---|---|
| Determ. | MSR | PU | Patch | 0.84 | 0.78 | 0.41 | 0.46 | 0.86 | 0.33 |
| | | | Thresh | 0.73 | 0.45 | 0.40 | 0.52 | 0.59 | - |
| | | | Image | - | - | - | - | - | 0.70 |
| TTD | PE | PU | Patch | 0.83 | 0.68 | 0.37 | 0.46 | 0.90 | 0.37 |
| | | | Thresh | 0.48 | 0.50 | 0.38 | 0.61 | 0.74 | - |
| | | | Image | - | - | - | - | - | 0.68 |
| | EE | AU | Patch | 0.74 | 0.69 | 0.36 | 0.43 | 0.90 | 0.37 |
| | | | Thresh | 0.53 | 0.53 | 0.29 | 0.40 | 0.88 | - |
| | | | Image | - | - | - | - | - | 0.68 |
| | MI | EU | Patch | 0.83 | 0.61 | 0.73 | 0.52 | 0.88 | 0.46 |
| | | | Thresh | 0.54 | 0.43 | 0.71 | 0.65 | 0.60 | - |
| | | | Image | - | - | - | - | - | 0.51 |
| Ensemble | PE | PU | Patch | 0.95 | 0.94 | 0.50 | 0.55 | 0.91 | 0.33 |
| | | | Thresh | 0.90 | 0.73 | 0.69 | 0.66 | 0.72 | - |
| | | | Image | - | - | - | - | - | 0.72 |
| | EE | AU | Patch | 0.94 | 0.83 | 0.44 | 0.49 | 0.89 | 0.29 |
| | | | Thresh | 0.78 | 0.19 | 0.12 | 0.53 | 0.53 | - |
| | | | Image | - | - | - | - | - | 0.67 |
| | MI | EU | Patch | 0.95 | 0.95 | 0.85 | 0.65 | 0.89 | 0.91 |
| | | | Thresh | 0.91 | 0.77 | 0.87 | 0.72 | 0.75 | - |
| | | | Image | - | - | - | - | - | 0.90 |
| TTA | PE | PU | Patch | 0.95 | 0.91 | 0.48 | 0.51 | 0.88 | 0.32 |
| | | | Thresh | 0.93 | 0.66 | 0.55 | 0.60 | 0.67 | - |
| | | | Image | - | - | - | - | - | 0.70 |
| | EE | AU | Patch | 0.95 | 0.83 | 0.44 | 0.46 | 0.87 | 0.29 |
| | | | Thresh | 0.89 | 0.27 | 0.16 | 0.49 | 0.53 | - |
| | | | Image | - | - | - | - | - | 0.67 |
| | MI | EU | Patch | 0.95 | 0.94 | 0.92 | 0.59 | 0.86 | 0.93 |
| | | | Thresh | 0.93 | 0.71 | 0.84 | 0.67 | 0.70 | - |
| | | | Image | - | - | - | - | - | 0.94 |
| SSN | PE | PU | Patch | 0.87 | 0.76 | 0.38 | 0.54 | 0.84 | 0.78 |
| | | | Thresh | 0.74 | 0.65 | 0.34 | 0.51 | 0.72 | - |
| | | | Image | - | - | - | - | - | 0.82 |
| | MI | AU | Patch | 0.68 | 0.63 | 0.32 | 0.54 | 0.72 | 0.53 |
| | | | Thresh | 0.67 | 0.51 | 0.25 | 0.49 | 0.57 | - |
| | | | Image | - | - | - | - | - | 0.55 |
| | EE | EU | Patch | 0.87 | 0.90 | 0.43 | 0.54 | 0.85 | 0.78 |
| | | | Thresh | 0.74 | 0.68 | 0.78 | 0.50 | 0.68 | - |
| | | | Image | - | - | - | - | - | 0.86 |

(b) AUROC scores

### G.3 QUALITATIVE RESULTS

In the following sections, samples are shown for the qualitative analysis to answer Q1 and Q2 from the separation study (see Sec. 4.4).

#### G.3.1 QUALITATIVE RESULTS FOR THE TOY DATASET

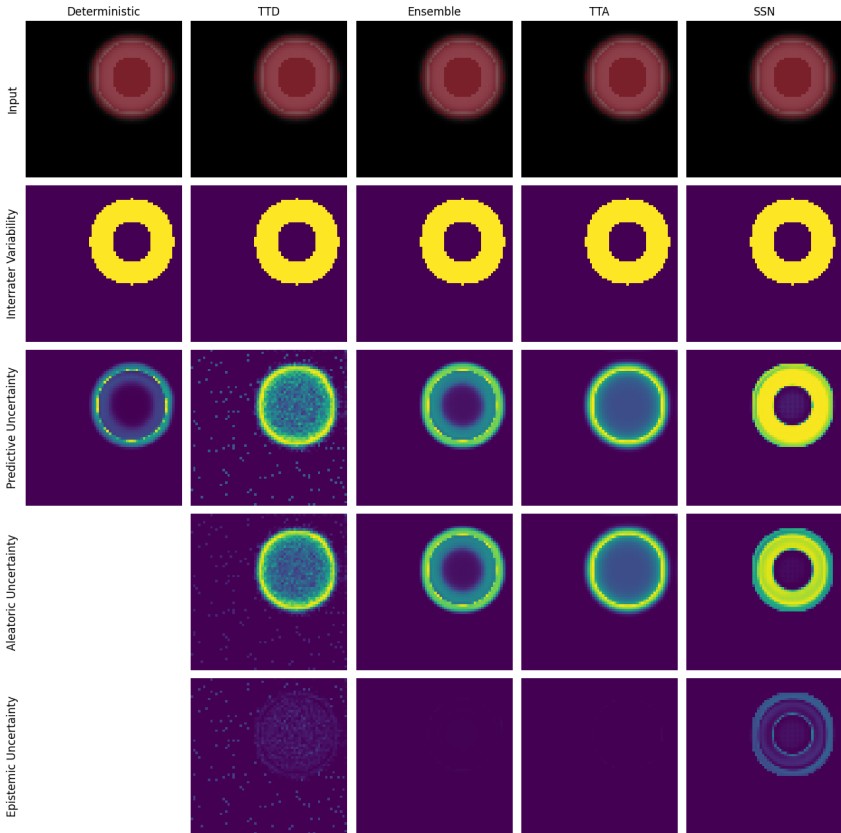

Figure 9: Qualitative results for separating aleatoric and epistemic uncertainty for the toy dataset. The reference segmentations are shown as overlay over the input image. Further, the interrater variability based on the pixel variance is shown. The uncertainty scores per pixel are normalized between $0$ and $0.5$ for the deterministic model and between $0$ and $0.7$ for the other prediction models, reflecting the possible range of uncertainty values.

### G.3.2 Qualitative results for the LIDC-IDRI datasets

Texture shift i.i.d. example

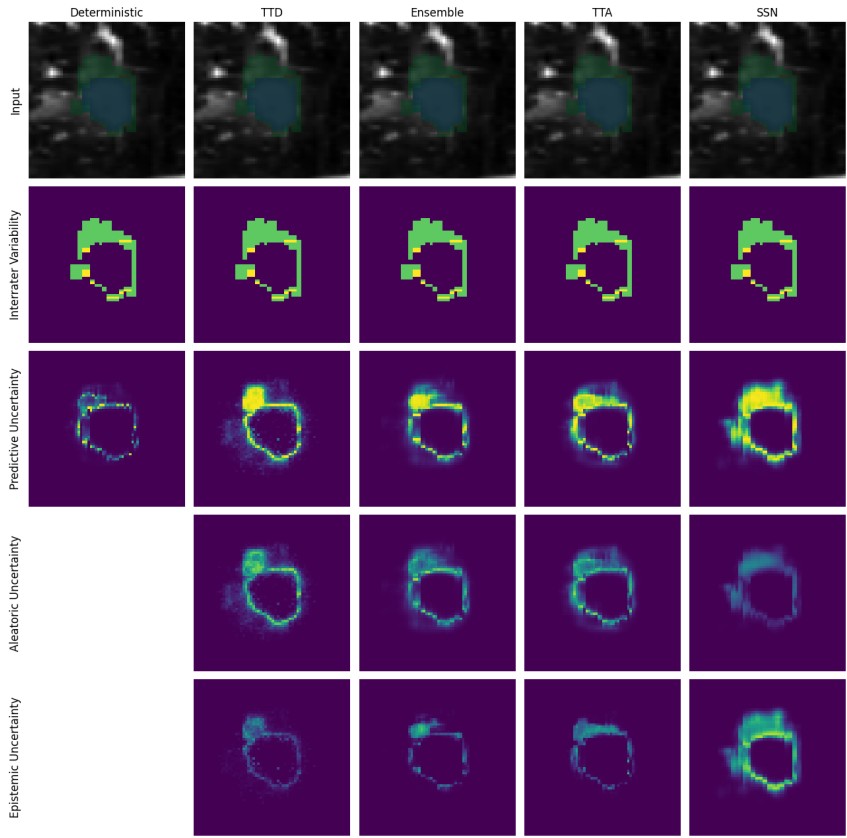

Figure 10: Qualitative results for separating aleatoric and epistemic uncertainty for the LIDC TEX dataset. A case that is part of the i.i.d. test set is shown. The reference segmentations are shown as overlay over the input image. Further, the interrater variability based on the pixel variance is shown. The uncertainty scores per pixel are normalized between 0 and 0.5 for the deterministic model and between 0 and 0.7 for the other prediction models, reflecting the possible range of uncertainty values.

Texture shift OoD example

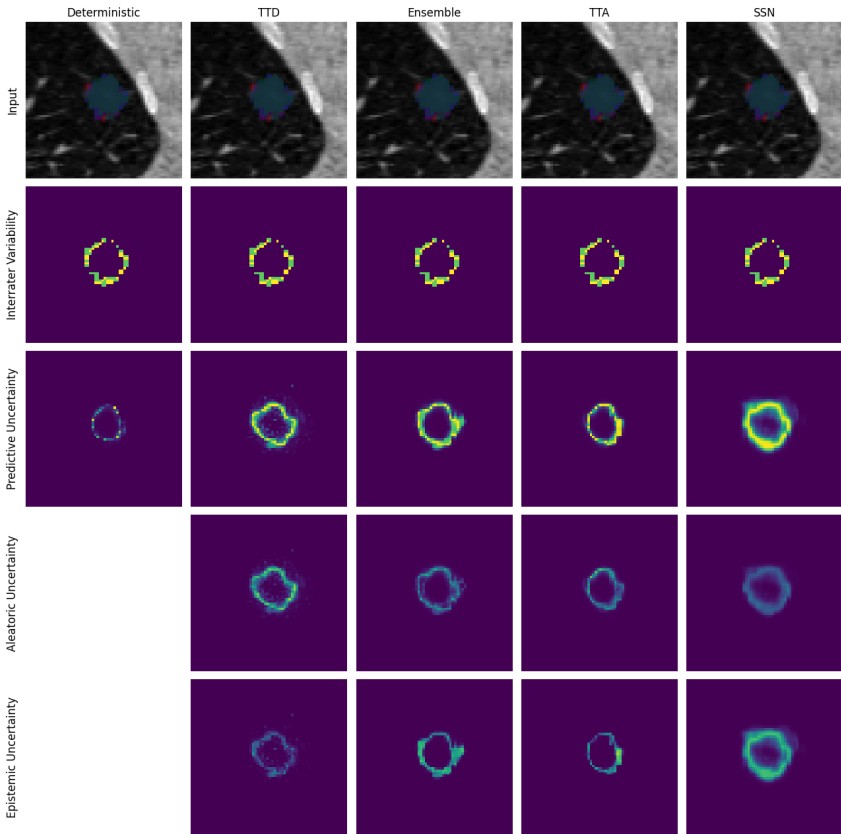

Figure 11: Qualitative results for separating aleatoric and epistemic uncertainty for the LIDC TEX dataset. A case that is part of the OoD test set is shown. The reference segmentations are shown as overlay over the input image. Further, the interrater variability based on the pixel variance is shown. The uncertainty scores per pixel are normalized between $0$ and $0.5$ for the deterministic model and between $0$ and $0.7$ for the other prediction models, reflecting the possible range of uncertainty values.

Malignancy shift i.i.d. example

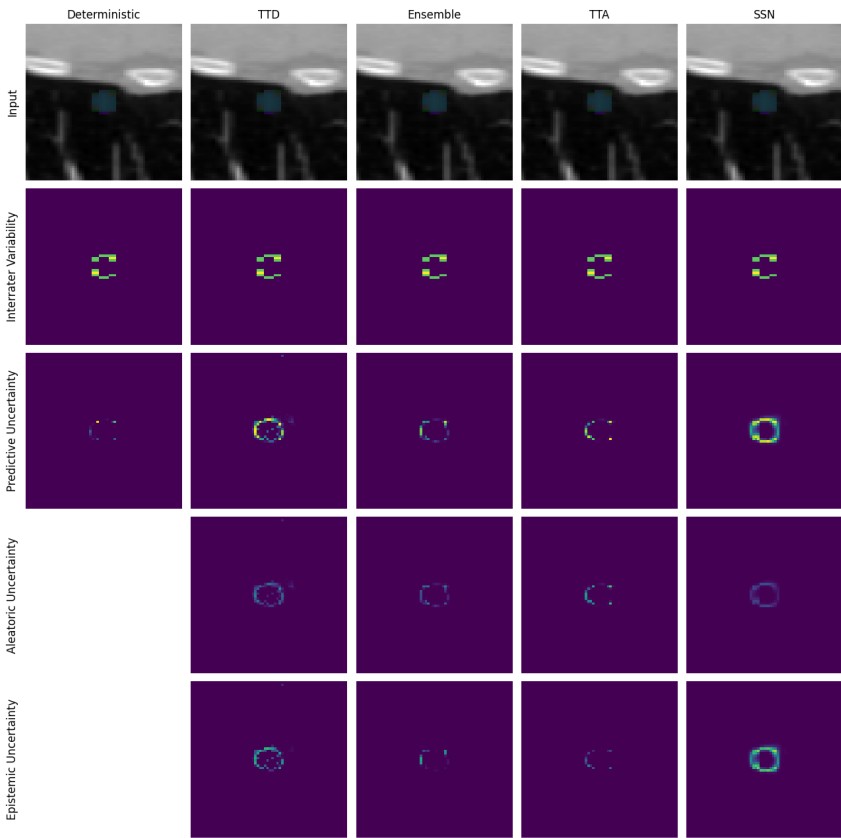

Figure 12: Qualitative results for separating aleatoric and epistemic uncertainty for the LIDC MAL dataset. A case that is part of the i.i.d. test set is shown. The reference segmentations are shown as overlay over the input image. Further, the interrater variability based on the pixel variance is shown. The uncertainty scores per pixel are normalized between $0$ and $0.5$ for the deterministic model and between $0$ and $0.7$ for the other prediction models, reflecting the possible range of uncertainty values.

Malignancy shift OoD example

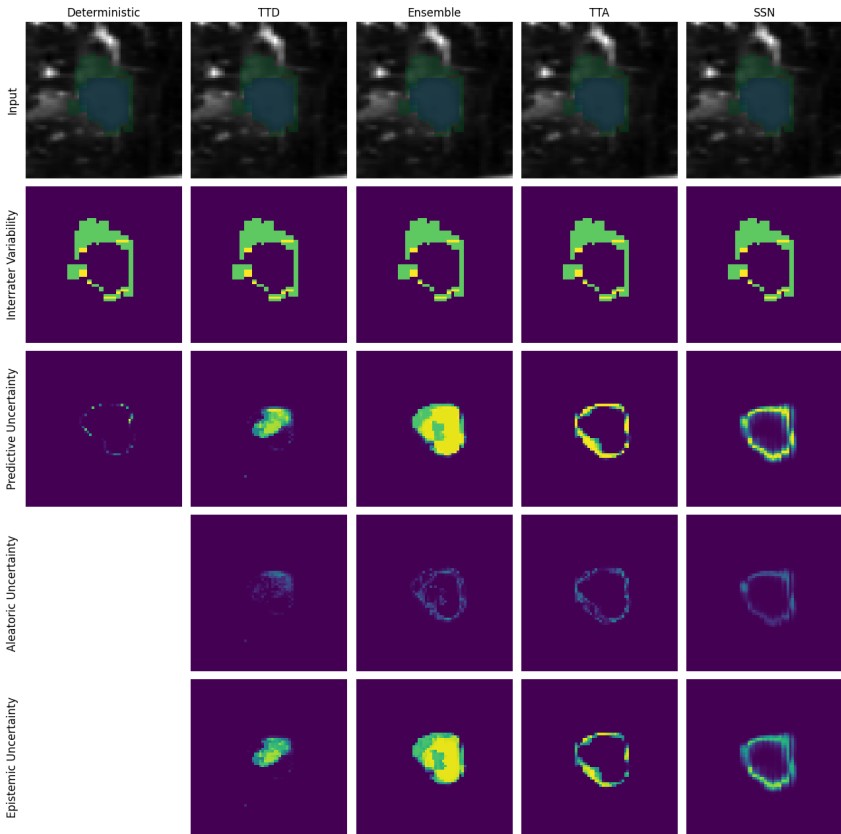

Figure 13: Qualitative results for separating aleatoric and epistemic uncertainty for the LIDC MAL dataset. A case that is part of the OoD test set is shown. The reference segmentations are shown as overlay over the input image. Further, the interrater variability based on the pixel variance is shown. The uncertainty scores per pixel are normalized between 0 and 0.5 for the deterministic model and between 0 and 0.7 for the other prediction models, reflecting the possible range of uncertainty values.

### G.3.3 QUALITATIVE RESULTS FOR THE GTA 5 / CITYSCAPES DATASET

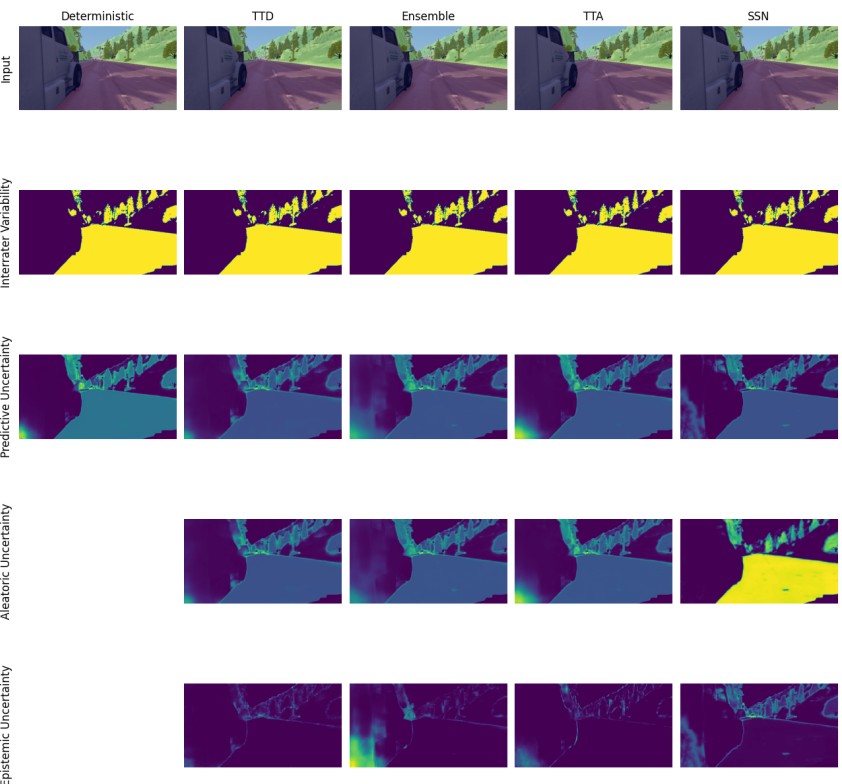

Figure 14: Qualitative results for separating aleatoric and epistemic uncertainty for the GTA 5 / Cityscapes dataset. The reference segmentations are shown as overlay over the input image. Further, the interrater variability based on the pixel variance is shown. The uncertainty scores per pixel are normalized per image.

## H  DETAILED RESULTS OF THE EVALUATION ON DOWNSTREAM TASKS

The following tables show the detailed results on the downstream tasks as described in Sec. 4.5. For the LIDC datasets, the results are shown in Table 5, while for the GTA5/CS dataset, the results are shown in Table 6.

Table 5: **Evaluation of downstream tasks on the LIDC datasets.** The table shows the segmentation performance by means of the Dice score and evaluation metrics for 5 different downstream tasks, where ↑ depict higher scores are better and ↓ lower scores are better. All scores are multiplied by $10^2$. The color heatmap is normalized per column and per shift, brighter columns imply better scores. For AL, the second cycle was only executed with EU and PU, indicated by empty grey entries for AU. Reported results show the mean and standard deviation over 3 different seeds. Abbreviations: PM: Prediction model, UM: Uncertainty measure, UT: Modeled uncertainty Type (according to theory), AGG: Aggregation strategy.

| | | | | | Seg. Performance | | OoD-D | Failure Detection | | | | AL | Calibration | | Ambiguity Modeling | | | |
|---|---|---|---|---|---|---|---|---|---|---|---|---|---|---|---|---|---|---|
| Shift | PM | UM | UT | AGG | Dice i.i.d. | Dice OoD↑ | AUROC↑ | AURC i.i.d.↓ | AURC OoD↓ | EAURC i.i.d.↓ | EAURC OoD↓ | Improv. OoD↑ | ACE i.i.d.↓ | ACE OoD↓ | NCC i.i.d.↑ | NCC OoD↑ | GED i.i.d.↓ | GED OoD↓ |

Table 6: **Evaluation of downstream tasks on the GTA 5 / Cityscapes dataset.** The table shows the segmentation performance by means of the Dice score and evaluation metrics for 5 different downstream tasks, where ↑ depict higher scores are better and ↓ lower scores are better. All scores are multiplyed by $10^2$. The color heatmap is normalized per column, brighter columns imply better scores. For AL, the second cycle was only executed with EU and PU, indicated by empty grey entries for AU. Reported results show the mean and standard deviation over 3 different seeds. Abbreviations: PM: Prediction model, UM: Uncertainty measure, UT: Modeled uncertainty Type (according to theory), AGG: Aggregation strategy.

| PM | UM | UT | AGG | Seg. Performance Dice i.i.d. ↑ | Seg. Performance Dice OoD ↑ | OoD-D AUROC ↑ | Failure Detection AURC i.i.d. ↓ | Failure Detection AURC OoD ↓ | Failure Detection E-AURC i.i.d. ↓ | Failure Detection E-AURC OoD ↓ | AL Improv. OoD ↑ | Calibration ACE i.i.d. ↓ | Calibration ACE OoD ↓ | Ambiguity Modeling NCC i.i.d. ↑ | Ambiguity Modeling NCC OoD ↑ | Ambiguity Modeling GED i.i.d. ↓ | Ambiguity Modeling GED OoD ↓ |
|---|---|---|---|---|---|---|---|---|---|---|---|---|---|---|---|---|---|
| Determ. | MSR | PU | Patch | 71.73±0.13 | 58.37±0.51 | 33.43±1.16 | 26.52±0.23 | 39.24±0.75 | 7.28±0.15 | 8.15±1.72 | -0.5±1.78 | 13.09±0.15 | 17.21±0.31 | 50.61±0.43 | 46.81±0.61 | 36.09±0.26 | 58.87±0.83 |
| Determ. | | | Image | 71.73±0.13 | 58.37±0.51 | 69.99±0.58 | 24.24±0.17 | 36.85±0.33 | 4.96±0.04 | 5.76±0.24 | 1.44±1.73 | 13.09±0.15 | 17.21±0.31 | 50.61±0.43 | 46.81±0.61 | 36.09±0.26 | 58.87±0.83 |
| TTD | PE | PU | Patch | 71.63±0.14 | 58.62±0.91 | 33.67±1.42 | 26.77±0.2 | 39.46±0.57 | 7.43±0.06 | 8.43±0.55 | -0.29±2.72 | 15.14±0.11 | 15.07±0.07 | 26.68±0.29 | 25.57±3.5 | 34.03±0.28 | 56.08±1.61 |
| TTD | | | Image | 71.63±0.14 | 58.62±0.91 | 67.8±0.71 | 24.51±0.21 | 36.14±0.3 | 5.16±0.06 | 5.12±0.09 | 0.94±2.81 | 15.14±0.11 | 15.07±0.07 | 26.68±0.29 | 25.57±3.5 | 34.03±0.28 | 56.08±1.61 |
| TTD | EE | AU | Patch | 71.63±0.14 | 58.62±0.91 | 36.83±1.35 | 26.76±0.21 | 39.45±0.61 | 7.41±0.07 | 8.43±0.58 | | 15.08±0.11 | 15.18±0.08 | 27.67±0.29 | 26.39±3.46 | 34.03±0.28 | 56.08±1.61 |
| TTD | | | Image | 71.63±0.14 | 58.62±0.91 | 68.02±0.75 | 24.5±0.21 | 36.15±0.3 | 5.15±0.06 | 5.13±0.09 | 0.03±2.86 | 15.08±0.11 | 15.18±0.08 | 27.67±0.29 | 26.39±3.46 | 34.03±0.28 | 56.08±1.61 |
| TTD | MI | EU | Patch | 71.63±0.14 | 58.62±0.91 | 46.34±4.04 | 26.88±0.11 | 39.18±0.6 | 7.45±0.06 | 8.15±0.46 | 0.45±2.82 | 17.56±0.13 | 17.82±0.5 | -23.17±0.19 | -12.85±2.61 | 34.03±0.28 | 56.08±1.61 |
| TTD | | | Image | 71.63±0.14 | 58.62±0.91 | 51.08±2.53 | 26.05±0.24 | 36.61±0.38 | 6.71±0.09 | 5.58±0.16 | | 17.56±0.13 | 17.82±0.5 | -23.17±0.19 | -12.85±2.61 | 34.03±0.28 | 56.08±1.61 |
| Ensemble | PE | PU | Patch | 71.92±0.14 | 59.36±0.46 | 32.72±0.57 | 26.4±0.13 | 38.33±0.63 | 7.42±0.07 | 8.08±0.56 | -0.65±1.67 | 15.64±0.12 | 15.61±0.22 | 23.77±1.11 | 25.37±0.98 | 31.92±0.22 | 50.12±0.36 |
| Ensemble | | | Image | 71.92±0.14 | 59.36±0.46 | 72.06±0.87 | 24.02±0.12 | 35.51±0.83 | 5.05±0.01 | 5.16±0.17 | 1.17±1.63 | 15.64±0.12 | 15.61±0.22 | 23.77±1.11 | 25.37±0.98 | 31.92±0.22 | 50.12±0.36 |
| Ensemble | EE | AU | Patch | 71.92±0.14 | 59.36±0.46 | 28.69±0.35 | 26.5±0.15 | 38.79±0.42 | 7.4±0.08 | 8.44±0.01 | | 15.57±0.11 | 15.3±0.21 | 26.74±0.14 | 29.59±0.9 | 31.92±0.22 | 50.12±0.36 |
| Ensemble | | | Image | 71.92±0.14 | 59.36±0.46 | 66.81±0.94 | 24.05±0.11 | 35.54±0.78 | 5.08±0.01 | 5.19±0.22 | | 15.57±0.11 | 15.3±0.21 | 26.74±0.14 | 29.59±0.9 | 31.92±0.22 | 50.12±0.36 |
| Ensemble | MI | EU | Patch | 71.92±0.14 | 59.36±0.46 | 90.63±0.29 | 26.69±0.04 | 38.57±1.15 | 7.22±0.13 | 8.22±1.61 | 1.77±1.79 | 21.33±0.06 | 17.43±0.28 | -22.55±0.08 | -6.06±0.79 | 31.92±0.22 | 50.12±0.36 |
| Ensemble | | | Image | 71.92±0.14 | 59.36±0.46 | 90.05±0.29 | 25.68±0.09 | 36.51±1.29 | 6.71±0.08 | 6.16±0.63 | 1.85±1.8 | 21.33±0.06 | 17.43±0.28 | -22.55±0.08 | -6.06±0.79 | 31.92±0.22 | 50.12±0.36 |
| VLL | PE | PU | Patch | 71.82±0.13 | 58.39±0.47 | 31.66±1.71 | 26.54±0.26 | 39.44±0.94 | 7.38±0.15 | 8.39±1.07 | -0.61±1.71 | 15.37±0.14 | 15.42±0.31 | 24.84±0.23 | 28.14±1.28 | 33.52±0.26 | 54.06±1.72 |
| VLL | | | Image | 71.82±0.13 | 58.39±0.47 | 69.81±0.32 | 24.27±0.18 | 36.44±0.16 | 5.11±0.05 | 5.39±0.05 | 1.44±1.78 | 15.37±0.14 | 15.42±0.31 | 24.84±0.23 | 28.14±1.28 | 33.52±0.26 | 54.06±1.72 |
| VLL | EE | AU | Patch | 71.82±0.13 | 58.39±0.47 | 28.85±0.18 | 26.55±0.27 | 39.35±0.98 | 7.39±0.17 | 8.35±1.12 | | 15.3±0.14 | 15.32±0.33 | 26.29±0.24 | 30.47±1.19 | 33.52±0.26 | 54.06±1.72 |
| VLL | | | Image | 71.82±0.13 | 58.39±0.47 | 66.61±0.31 | 24.28±0.18 | 36.42±0.16 | 5.12±0.05 | 5.37±0.05 | 2.12±1.77 | 15.3±0.14 | 15.32±0.33 | 26.29±0.24 | 30.47±1.19 | 33.52±0.26 | 54.06±1.72 |
| VLL | MI | EU | Patch | 71.82±0.13 | 58.39±0.47 | 92.75±0.11 | 27.1±0.09 | 39.65±0.47 | 7.05±0.08 | 8.61±0.41 | 2.12±1.77 | 22.04±0.39 | 19.1±0.32 | -16.25±0.14 | -4.45±1.2 | 33.52±0.26 | 54.06±1.72 |
| VLL | | | Image | 71.82±0.13 | 58.39±0.47 | 94.32±0.33 | 26.12±0.05 | 37.57±0.41 | 6.96±0.08 | 6.53±0.54 | 2.46±1.77 | 22.04±0.39 | 19.1±0.32 | -16.25±0.14 | -4.45±1.2 | 33.52±0.26 | 54.06±1.72 |
| NSS | PE | PU | Patch | 65.49±0.13 | 51.9±0.44 | 77.95±1.28 | 33.85±0.35 | 46.53±0.89 | 10.14±0.29 | 9.91±0.55 | 2.01±2.21 | 16.1±0.13 | 17.84±0.41 | 55.97±0.09 | 37.26±0.95 | 25.6±0.03 | 45.26±1.1 |
| NSS | | | Image | 65.49±0.13 | 51.9±0.44 | 82.0±1.1 | 30.07±0.05 | 43.05±0.95 | 6.43±0.41 | 6.43±0.41 | 3.21±2.25 | 16.1±0.13 | 17.84±0.41 | 55.97±0.09 | 37.26±0.95 | 25.6±0.03 | 45.26±1.1 |
| NSS | MI | AU | Patch | 65.49±0.13 | 51.9±0.44 | 53.1±3.37 | 32.57±0.31 | 46.28±1.29 | 8.86±0.4 | 9.65±1.0 | | 21.6±0.14 | 25.8±0.42 | 70.34±0.36 | 51.8±0.44 | 25.6±0.03 | 45.26±1.1 |
| NSS | | | Image | 65.49±0.13 | 51.9±0.44 | 55.21±1.37 | 31.0±0.09 | 46.07±0.67 | 7.29±0.13 | 9.44±0.51 | 2.44±2.17 | 21.6±0.14 | 25.8±0.42 | 70.34±0.36 | 51.8±0.44 | 25.6±0.03 | 45.26±1.1 |
| NSS | EE | EU | Patch | 65.49±0.13 | 51.9±0.44 | 78.12±1.74 | 33.87±0.42 | 46.86±0.86 | 10.16±1.37 | 7.04±0.56 | | 10.49±0.31 | 11.12±0.45 | 4.57±1.07 | 2.95±0.9 | 25.6±0.03 | 45.26±1.1 |
| NSS | | | Image | 65.49±0.13 | 51.9±0.44 | 86.2±1.08 | 32.58±0.43 | 43.62±1.1 | 8.87±0.41 | 7.0±0.56 | 2.37±1.91 | 10.49±0.31 | 11.12±0.45 | 4.57±1.07 | 2.95±0.9 | 25.6±0.03 | 45.26±1.1 |

