# OpenReview forum: "ValUES: A Framework for Systematic Validation of Uncertainty Estimation in Semantic Segmentation"
_ICLR.cc/2024/Conference — ICLR 2024 oral_

### Official Review · Reviewer_u7mE · 2023-10-23

**Soundness:** 4 excellent
**Presentation:** 4 excellent
**Contribution:** 4 excellent
**Rating:** 8
**Confidence:** 4

**Summary:**

This paper addresses the widely recognized disparity between theory and practical implementation within the field of uncertainty estimation in semantic segmentation. In order to bridge this gap, the paper highlights the three key components of any uncertainty method and outlines the associated pitfalls and requirements for addressing these pitfalls.

**Strengths:**

The paper's approach to addressing the gap between theory and practice in the field is commendable and serves to underscore its originality and significance. It offers a systematic framework that effectively addresses limitations seen in prior research. The paper appropriately points out the various pitfalls present in earlier works, even though they were generally well-regarded (some having high citation counts). To the best of my knowledge, this study represents the first attempt to systematically and explicitly evaluate these pitfalls across not only a toy dataset but also two real-world datasets (LIDC-IDRI and GTA5/Cityscapes). The insights provided by the authors are valuable in confirming that certain theoretical claims, originally validated on toy datasets, may not necessarily translate to at least two real-world datasets. Moreover, it sheds light on what works effectively with ensembles, a commonly used benchmark for uncertainty estimation.

I have to highlight that in this type of works it is always possible to think about more architectures, more datasets and more analysis, but given the limited space (in terms of pages) available, and all the material already in appendices, I believe that the authors chose a good scope for their analysis. In this sense, the quality and clarity of this work is equally very good, given how difficult it is to summarise so many results - Figure 2 (with more details in appendix) thus seemed to me an acceptable and useful way to present such results.

This work will likely become essential for anyone in the field seeking scientific advancement and a deeper understanding of uncertainty estimations within their specific applications. Hopefully, this paper will prompt the community to pay closer attention to their validation practices. In light of its potential for significant impact in multiple areas, I confidently endorse for its acceptance. I believe it would be fair for this work to have a score of 10, but I prefer to wait for the remaining reviews, during the rebuttal period.

**Weaknesses:**

It is regrettable that component C0 was fixed (for each dataset) and not explored for at least another architecture. However, I might say that such a weakness is well within the realm of usual limitations expected in a paper.

The decisions pertaining to component C1 involve numerous hyperparameters, with some known to have a significant impact on performance. Consequently, this represents another weakness in the study, as it appears that these hyperparameters were not thoroughly investigated.

Figure 2 uses different colours to distinguish distinct parts of the figure. This is not friendly for people with colour deficiency, and the authors could try to include patterns on top of the colour scheme.

**Questions:**

I do not have any further questions.

---

> ### Author Response · Authors · 2023-11-17
>
> Thank you again for your valuable comments, and for taking the time to read our general reply, as well as considering our point-by-point comments here:
>
> ---
> W1. It is regrettable that component C0 was fixed (for each dataset) and not explored for at least another architecture. However, I might say that such a weakness is well within the realm of usual limitations expected in a paper.
>
> - We appreciate your reflective feedback, and agree with you that ablating different segmentation backbones would be an interesting extension. However, as you agree, given the extent of the provided experiments, we had to restrict ablations at some point.
>
> ---
> W2. The decisions pertaining to component C1 involve numerous hyperparameters, with some known to have a significant impact on performance. Consequently, this represents another weakness in the study, as it appears that these hyperparameters were not thoroughly investigated.
>
> - Thank you for pointing this out. We are aware that selecting the right hyperparameters can influence the results for the different prediction models.
> - Therefore, we state the possible impact of hyperparameters in Appendix C (“It is important to note that we did not do extensive hyperparameter searches for all the hyperparameters that we state here but rather used standard values if they worked reasonably and if not, we performed sweeps over the hyperparameters to find appropriate settings.”).
> - However, in response to your feedback and to make the potential effects of hyperparamteres even more prominent in the main paper, we added a respective paragraph in our uploaded revision in Sec. 4.3., where we also provide direct links to respective hyperparameter specifications in Appendix C.
>
> ---
> W3. Figure 2 uses different colours to distinguish distinct parts of the figure. This is not friendly for people with colour deficiency, and the authors could try to include patterns on top of the colour scheme.
>
> - Thank you for making this important point. To make the barplots in Fig. 2 and Fig. 3 (formerly Fig. 2) more accessible, we added patterns on top of the bars that help to distinguish the categories as suggested.
>
> ---
> Thank you once more for your feedback and for highlighting the importance of this kind of reflective work and its potential benefits for the community. Following up on your statement regarding the scoring, please let us know in case there are any remaining concerns that would hinder you from giving a higher score.

---

> > ### Comment · Reviewer_u7mE · 2023-11-20
> >
> > I thank the authors for addressing the weaknesses I have identified in this work. We seem to more or less agree that they are indeed weaknesses of this work, and that they are inline with typical scope limitations of conference papers. I will wait for the remaining reviewers to answer the authors before making a final decision on my scores.

---

> > > ### Author Response · Authors · 2023-11-23
> > > **Thank you for your response**
> > >
> > > Thank you very much for taking the time to read our response. We are looking forward to your decision.

---

### Official Review · Reviewer_6vjf · 2023-10-30

**Soundness:** 3 good
**Presentation:** 2 fair
**Contribution:** 3 good
**Rating:** 8
**Confidence:** 4

**Summary:**

Authors present an empirical study to assess uncertainty estimation
methods, focusing on their ability to separate aleatoric and epistemic
uncertainty and performance on down-stream tasks. Through three data
sets, one being completely toy. From the experimental results, author
present their conclusions in the main text. Methods cannot seem to
separate AU from EU in real data sets. Performance on downstream tasks
vary greatly.

**Strengths:**

1. Uncertainty estimation is an important problem. Assessment of
   estimated uncertainty is a challenging yet critical topic.
2. Evaluation through multiple downstream tasks is an interesting
   direction. Adopting this approach would surely improve the
   assessment methodology in the field. This is possibly the strongest
   point of this work.
3. Authors seem to have done a large amount of experimentation
   covering different aspects.

**Weaknesses:**

I have two concerns regarding this article:

1. I am not sharing the enthusiasm of the authors regarding the need
   to separate AU and EU. They note that downstream tasks are
   important - which I fully agree. To solve those downstream tasks, I
   would not require a separation of uncertainty terms. What is the
   main value of this separation beyond a "theoretical" understanding?
   Perhaps authors should provide a strong justification as to why one
   must care about this separation...
2. I find the presentation style not optimal. Authors mostly provide
   summary results in the main text and the quantitative results in
   the appendix. Appendix should be there to support an otherwise
   self-contained main text. Here, I do not think the main text is
   self-contained.

**Questions:**

1. What is the aim for separating AU from EU?

---

> ### Author Response · Authors · 2023-11-17
>
> Thank you again for your valuable comments and for taking the time to read our general reply, as well as considering our point-by-point comments here:
>
> ---
> W1. I am not sharing the enthusiasm of the authors regarding the need to separate AU and EU. [..] Perhaps authors should provide a strong justification as to why one must care about this separation…
> - Thank you for bringing up this important point. We think this is a misunderstanding because we totally agree with you: we are not enthusiastic about separation, in contrast, we take the disconnect between the prevalent enthusiasm and a lack of evidence as our motivation to challenge the separation by asking
>   1. Is it feasible? We investigate this through our separation study and check whether the proposed methods actually perform the separation that they are stated to perform.
>   2. Is it important? We systematically test on various downstream tasks whether isolating uncertainty types brings actual benefits.
> - Since numerous papers, some of which are highly cited, are based on the assumptions that separation is feasible and beneficial without providing convincing evidence for either of the two, we hope you agree that scrutinizing the current enthusiasm in the field by means of this study is an important contribution.
> - That said, in some downstream tasks like AL or OoD-detection, there is a theoretic case that isolating samples featuring epistemic uncertainty is preferable to a general predictive uncertainty containing a measure for ambiguities in the data.
> - In Fig. 3 (formerly Fig. 2), in the column “uncertainty measure”, we provide empirical evidence indicating that for some tasks like OoD detection or ambiguity modeling, the isolation of specific uncertainty types seems beneficial, but for others, such as calibration, it is not.
> - In response to your valuable feedback we now, 1) updated the introduction to highlight our objective stance and state a need to question the feasibility and importance of separation. 2) We updated the respective conclusion (Sec. 5 “R1”) to now state a clear demand for providing evidence about the feasibility and benefits of separation in future research.
> - We hope our study and benchmark can help to foster more evidence-based research in this field in the future.
> ---
> W2. I find the presentation style not optimal. Authors mostly provide summary results in the main text and the quantitative results in the appendix. Appendix should be there to support an otherwise self-contained main text. Here, I do not think the main text is self-contained.
> - Thank you for this very helpful comment.
> - To clarify, the main results of Sec. 4.5, where we benchmark prevalent methods on a wide variety of downstream tasks, are fully contained in Fig. 3 (formerly Fig. 2).
> - Regarding Sec. 4.4 (“separation study”), please see the detailed response in our general reply above. In the revision, we provide all results in the main paper and clearly indicate where to find them, ensuring the self-containment of the main paper.
> ---
> Q1. What is the aim for separating AU from EU?
> - Please see W1 for a detailed answer.
> ---
> Thank you once more for your constructive feedback. As we believe to have resolved your comments, please let us know if there are any remaining concerns that would hinder a recommendation for acceptance.

---

> > ### Comment · Reviewer_6vjf · 2023-11-22
> > **Thank you for the clarifications**
> >
> > I thank the author for the response. I very much appreciate the changes and also the critical view authors present.

---

> > > ### Author Response · Authors · 2023-11-22
> > > **Thank you for your response**
> > >
> > > Thank you very much for taking the time to read our response and re-considering your assessment based on the provided updates.

---

### Official Review · Reviewer_T66t · 2023-10-31

**Soundness:** 2 fair
**Presentation:** 3 good
**Contribution:** 2 fair
**Rating:** 6
**Confidence:** 4

**Summary:**

This paper investigates the gap between theory and practice in uncertainty estimation. It introduces an evaluation framework that provides a controlled setting for studying data ambiguities, method component ablations, and test environments for various uncertainty applications.

**Strengths:**

**	The paper is easy to follow and well-organized.

**	The research direction of this article is compelling

**	The experimental results are pretty strong.

**Weaknesses:**

**  The novelty is limited. The primary contribution seems to conduct additional experiments using existing methods.

**  The experiments presented in the main text is not persuasiveness, and the binary (yes or no) outcomes remain inconclusive.

**  Please provide detailed analytically experimental or theoretical proofs

**  It would be better to conduct more ablation studies using different backbones (e.g., VIT based model).

**Questions:**

**  What is the main innovation of this paper? Please provide detailed explanations of the contribution points.

**  Please show the justification to establish the significance of the motivation. It is suggested to explain why this motivation is particularly important.

**  For Q1-Q4 questions, please supply comprehensive quantitative data analyses..

---

> ### Author Response · Authors · 2023-11-17
>
> Thank you again for your time to review our work. There might be misconceptions regarding the scope of ICLR as well as the provision of experiments. We believe our reply below resolves the related concerns.
>
> ---
> W1. The novelty is limited. The primary contribution seems to conduct additional experiments using existing methods.
> - We think there is a misunderstanding about the scope of ICLR. We specifically submitted to the primary area “datasets and benchmarks”, a dedicated area where the novelty does not lie in new methods.
> - The legitimacy is also evident in numerous ICLR papers sharing our format and type of contribution: E.g., Jaeger et al. ICLR23 (Top 5%), Galil et al. ICLR23a (Top 25%), Galil et al. ICLR 2023b, Wiles et al. ICLR 2022 (oral), Zong et al. ICLR23 (Top 25%).
> - **Motivation**: In contrast, the exaggerated emphasis on methodological novelty is a driving factor behind the problems of current literature we identify in our work (see Sec. 3). The resulting inconsistencies call for a reflective and rigorous understanding of existing methods.
> - **Novelty**: our study is the first to:
>   - systematically reveal profound shortcomings in the practices of uncertainty estimation for segmentation. E.g. we identify 1) contradictions about the feasibility of separating uncertainty types, 2) a disconnection between theoretic work and concrete downstream applications, 3) and a widespread neglection of essential components.
>   - present a framework that is rigorously designed to overcome the pitfalls by following concretely stated requirements for meaningful benchmarking.
>   - conduct a large-scale empirical study that compares prevalent methods on an unprecedented spectrum of data sets, tasks, and uncertainty settings - generating novel insights.
> - **Concrete impact**:
>   - prior contradictions in literature are resolved, fostering a consistent picture, e.g. the ability of TTA to model epistemic uncertainty or the importance of score aggregation strategies.
>   - methodological researchers are equipped with an easy-to-use framework to test their methods against potent baselines and under meaningful settings.
>   - practitioners can access a knowledge base enabling informed decisions about the appropriate uncertainty method for their task.
> ---
> W2. The experiments [..] in the main text is not persuasiveness, [...].
> - To clarify, the main results of Sec. 4.5, where we benchmark methods on downstream tasks, are fully contained in Fig. 3 (formerly Fig. 2).
> - Regarding Sec. 4.4 (“separation study” / “Q1-Q4”), please see the detailed response in our general reply above. In the revision, we provide all results in the main paper and clearly indicate where to find them.
> - In general, all reviewers acknowledge our extensive experiments: “thoughtful experiments to study AU/EU”, “large amount of experimentation covering different aspects”, “this study represents the first attempt to systematically [..] evaluate these pitfalls across not only a toy dataset but also two real-world datasets”. This includes your comment: “The experimental results are pretty strong.”
> ---
> W3. Please provide detailed analytically experimental or theoretical proofs
> - Regarding experiments, we refer to our reply in W2.
> - Regarding theoretical proofs, where necessary, we provide theoretical derivations (see Appendix D and E).
> ---
> W4. It would be better to conduct more ablation studies using different backbones (e.g., VIT [..]).
> - Thank you for pointing this out. We agree that studying backbones would be an interesting extension.
> - However, given the extent of our work, we have to cut-off on ablations at some point. Or, as reviewer u7mE excellently phrases it: “I have to highlight that in this type of works it is always possible to think about more architectures, more datasets and more analysis, but [...] the authors chose a good scope for their analysis.”; “It is regrettable that component C0 was fixed [...] However, [..] such a weakness is well within the realm of usual limitations expected in a paper.”
> - Further, We argue that ViT is no general new SotA architecture in segmentation that would in some way diminish our CNN-based results. Instead, CNNs are still competitive [a,b] and the clear SotA in many domains, e.g. [c].
>
> [a] Wang et al. “InternImage: Exploring Large-Scale Vision Foundation Models with Deformable Convolutions”\
> [b] Liu et al. “A ConvNet for the 2020s”\
> [c] Isensee et al. “nnU-Net: a self-configuring method for deep learning-based biomedical image segmentation”
>
> ---
> Q1. What is the main innovation[..]? [..]
> - Please see our response to W1.
> ---
> Q2. Please [..] explain why this motivation is [..] important.
> - Please see our response to W1.
> ---
> Q3. For Q1-Q4 questions, please supply [..] analyses.
> - Please see our response to W2.
> ---
> Thank you for your constructive feedback. As we believe to have resolved your comments, please let us know if there are any remaining concerns that would hinder a recommendation for acceptance.

---

> ### Author Response · Authors · 2023-11-23
>
> We thank the reviewer again for taking the time to review our paper. However, it is unfortunate that there was no acknowledgment of our response and, thus, no resulting discussion. This is especially noteworthy, as we believe to have resolved the concerns stated in this review, and the respective score of 3 is a clear outlier among all reviews.

---

### Official Review · Reviewer_NXFz · 2023-11-01

**Soundness:** 4 excellent
**Presentation:** 4 excellent
**Contribution:** 4 excellent
**Rating:** 8
**Confidence:** 4

**Summary:**

In this work, the authors provide an extensive study and guidelines for uncertainty estimation (UE) and its applications. Specifically, they assign terminologies to different components in the UE pipeline and study how each component affects the end result. UE as a field has several ambiguous and poorly proven results, and hence the authors’ work provides a meaningful understanding. They provide extensive empirical results to validate their claims.

**Strengths:**

1) The authors ask several good questions such as: separability of aleatoric and epistemic uncertainty; which predictive model and uncertainty measure to use; as well as how choosing the downstream task is important in designing a good UE pipeline.
2) The authors break down the whole UE pipeline and identify different components well. They also consider the most relevant downstream tasks for UE.
3) The authors provide a good discussion about existing pitfalls in UE research with respect to each component. They cite good references for this.
4) The authors conduct thoughtful experiments to study AU/EU, for eg: applying gaussian blur to the toy example and getting multiple raters’ annotations; changing shape/intensity for EU etc.

**Weaknesses:**

1) The authors mention that “Overall, surpassing the ‘random’ AL baseline appears challenging, with marginal improvements on LIDC MAL and GTA5/CS”. However, great strides have been made in AL, and works have obtained significant improvements like in [1]. Could the authors discuss this?
2) The authors mention that “The choice of aggregation method exhibits dataset-dependent variability” and “The choice of aggregation method yields mixed results on LIDC TEX, similar to other downstream tasks”. Is there any recommended guidelines on which approach works best in which setting, or, should users try all combinations?

**References**

[1] Wang, Wenzhe, et al. "Nodule-plus R-CNN and deep self-paced active learning for 3D instance segmentation of pulmonary nodules." Ieee Access 7 (2019): 128796-128805.

**Questions:**

1) Please also see weakness above.
2) It seems the authors have considered sampling-based methods in their study. How would the discussion be if the method is sampling-free like [2]?
3) As the authors have considered SSNs, would the discussion for probabilistic methods like Probabilistic UNet [3] be similar or different?

**References**

[2] Postels, Janis, et al. "Sampling-free epistemic uncertainty estimation using approximated variance propagation." Proceedings of the IEEE/CVF International Conference on Computer Vision. 2019.

[3] Kohl, Simon, et al. "A probabilistic u-net for segmentation of ambiguous images." Advances in neural information processing systems 31 (2018).

---

> ### Author Response · Authors · 2023-11-17
>
> Thank you again for your valuable comments, and for taking the time to read our general reply, as well as considering our point-by-point comments here:
>
> ---
> W1. The authors mention that “Overall, surpassing the ‘random’ AL baseline appears challenging [...]. However, [...] works have obtained significant improvements like in [1]. [...]
> - We agree that Deep AL is improving and there are more advanced strategies that do not only rely on uncertainty, but also diversity [1, a] or different training schemes [1].
> - However, the consensus is that outperforming the “random” baseline consistently is still challenging for both segmentation (“state-of-the-art active learning approaches often fail to outperform simple random sampling…” [b], [c Tab.6] shows that “random” performs best given small labeled datasets)  and classification (see “In contrast to previous findings, well-regularized random baseline in our study was either at par with or marginally inferior to the sota AL methods.” [d], and sub-random performance of multiple AL methods in [e Fig. 2]).
> - Similarly, in [1], there is no direct comparison to a “random” baseline with an identical starting budget and no standard deviations are given, making it hard to assess the performance improvement obtained with AL [1 Fig. 5].
> - In response to your feedback, we added a paragraph with references supporting our statement.
>
> [a] Sreenivasaiah et al. "Meal: Manifold embedding-based active learning."\
> [b] Mittal, et al. "Parting with illusions about deep active learning."\
> [c] Mittal et al. "Best Practices in Active Learning for Semantic Segmentation."\
> [d] Munjal et al. "Towards robust and reproducible active learning using neural networks."\
> [e] Lüth et al. "Navigating the Pitfalls of Active Learning Evaluation: A Systematic Framework for Meaningful Performance Assessment."
>
> ---
> W2. The authors mention that “The choice of aggregation method exhibits dataset-dependent variability” and “[...] yields mixed results [...]”. Is there any recommended guidelines [...], or, should users try all combinations?
> - While we have some working hypothesis on the pros and cons of applied aggregation strategies, there is no clear signal in our empirical results, thus, given testing the limited number of potential strategies is a cheap post-hoc analysis, our recommendation for reliable performance is empirical testing.
> - One advice for AL tasks could be to test the performance of aggregation strategies on proxy tasks before relying on a single strategy throughout the expensive AL training cycle.
> - We did identify one “red flag”: using sum-aggregation when there is a potential correlation with the object size in the image. Thus, another recommendation is to reflect on such correlations when choosing an aggregation strategy.
> - In response to your feedback, and to communicate these recommendations more prominently, we added a paragraph in Sec. 4.5.
>
> ---
> Q1. It seems the authors have considered sampling-based methods [...]. How would the discussion be if the method is sampling-free like [2]?
> - Since the method in [2] is proposed as an alternative to Monte Carlo-Dropout (MCD), and it is further compared to MCD showing similar behavior, we would assume that the behavior in our setting would be similar as well.
> - Evidence for this hypothesis is also given in [2 Fig.5], where the structural similarity of the uncertainty maps between Bayesian SegNet and their sampling-free method is shown.
> - It might also be interesting to discuss the performance of the uncertainty method in conjunction with the computational overhead, similarly to the cost-benefit tradeoff between ensembles and TTA discussed in our manuscript.
>
> ---
> Q2. As the authors have considered SSNs, would the discussion for probabilistic (Prob.) methods like Prob. UNet [3] be similar or different?
> - Both methods use random variables to model the spatially coherent distribution of outcomes given a sample (while the SSN does so by assuming a multivariate Gaussian over the distribution of logits, the Prob. U-Net utilizes a latent variable model during the forward pass). Therefore their behavior is expected to be similar.
> - However, SSNs are a more lightweight alternative to the Prob. U-Net (no separate network to estimate the distribution, no variational inference training), rendering SSNs more applicable for users, which is why we preferred this method in our benchmark.
> - In the SSN paper, the authors claim to outperform the Prob. U-Net in terms of GED. However, in theory, the Prob. U-Net is more expressive in its abilities to model ambiguity. This can be seen in [f Tab. 2] when both models are conditioned on modeling a specific style.
>
> [f] Zepf et al. “That Label's Got Style: Handling Label Style Bias for Uncertain Image Segmentation”
>
> ---
> Thank you once more for your constructive feedback. As we believe to have resolved your comments, please let us know in case you have remaining suggestions to further increase the quality of our work.

---

> > ### Comment · Reviewer_NXFz · 2023-11-21
> >
> > I thank the authors for responding to my questions. I believe this is a good paper and hence maintain my score.

---

> > > ### Author Response · Authors · 2023-11-22
> > > **Thank you for your response**
> > >
> > > Thank you very much for taking the time to read our response.

---

### Author Response · Authors · 2023-11-17

We sincerely thank all reviewers for their valuable comments. The reviewers generally agreed on the added value of our work (“This work will likely become essential for anyone in the field seeking scientific advancement”, “Adopting this approach would surely improve the assessment methodology in the field.”, “potential for significant impact in multiple areas”, “experimental results are pretty strong”).


Next to the poin-by-point responses, we will address the two main concerns here:


---
**Concern 1: Missing results in Section 4.4.** The first concern was the placement of the results table for our separation study (Sec. 4.4) in the appendix (raised by T66t and 6vjf).
- Thank you for pointing out the fact that the main paper was not self-contained in Sec. 4.4.
- To clarify, the main results of Sec. 4.5, where we benchmark prevalent methods on a wide variety of downstream tasks are fully contained in the main paper Fig. 3 (formerly Fig. 2).
- As for the uncertainty separation study in Sec. 4.4, in the uploaded revision we now provide all results in the main paper and make them quickly accessible: For the toy data set, results are in the new Fig. 2b), and for the LIDC and GTA5/CS data sets we now clearly indicate in the text as well as with gray-shaded icons, in which specific panels of  Fig. 3 (formerly Fig. 2) they are displayed.
- With this, the main paper is now fully self-contained.
- We further provide a stronger link now to the corresponding Appendix G which provides detailed non-aggregated results, a more detailed description of the study, as well as an accompanying qualitative analysis of uncertainty maps.

---
**Concern 2: “Limited novelty”.** T66t states of notion of “limited novelty” because “the primary contribution seems to conduct additional experiments using existing methods.”
- As detailed in the point-by-point response to reviewer T66t, we strongly disagree with this notion. To give an overview over our response:
- We think there is a misunderstanding about the scope of ICLR. We specifically submitted to the primary area “datasets and benchmarks”, a dedicated area where the novelty does not lie in new methods, but in either new datasets or a new benchmark comparing existing methods to generate new insights.
- The legitimacy is also evident in numerous ICLR papers sharing our format and type of contribution: E.g., Jaeger et al. ICLR23 (Top 5%), Galil et al. ICLR23a (Top 25%), Galil et al. ICLR 2023b, Wiles et al. ICLR 2022 (oral), Zong et al. ICLR23 (Top 25%).
- In contrast, we argue that the exaggerated emphasis on methodological novelty is a driving factor behind the problems of current literature we identify in our work (see Sec. 3). The resulting inconsistencies call for a reflective and rigorous understanding of existing methods. This is precisely the motivation of our work.
- In this motivation, we follow previous calls for reflection in the field such as “Novelty in Science” by Michael J. Black, or “Troubling Trends in Machine Learning Scholarship” by Lipton et al.
- In the detailed response, we go on to list the main novelties of our work and state the concrete benefits it brings to the community.

---
We have thoroughly revised our manuscript to address the provided feedback and uploaded the revision, where we highlight the modified parts in red. Changes include:
- We added results for the separation study in the main paper: For the toy dataset, we added the new Fig. 2b), for the real-world data sets, we clearly indicate the specific panels in Fig. 3 (formerly Fig. 2) displaying the results. Updated text descriptions further clarify the location of all results required to read the section, including a stronger link to Appendix G.
- We updated the introduction and conclusion to state a clear need for validating the importance and feasibility of separating uncertainty types.
- We set our conclusion w.r.t. AL into context with existing large-scale AL studies.
- We give concrete recommendations for how to select aggregation strategies in practice.
- We adapted Fig. 3 (formerly Fig. 2) to make it more accessible for people with color deficiency.

We also added a link to an anonymized GitHub repo including the code for reproducibility

---
We believe these updates resolve the stated concerns. Please find our point-by-point answers in the respective reviewer sections.

---

### Meta-Review · Area_Chair_J23q · 2023-12-05

**Metareview:**

This paper addresses the current gaps in the validation of segmentation uncertainty. As the authors point out, this is a research field with a large number of empirical problems, including lack of benchmarks, lack of well-defined tasks, lack of well-defined metrics for validation. This paper makes a thorough contribution towards sound validation procedures.

The main weaknesses of the paper related to presentation and semantics, and have been addressed by the authors in the discussion phase. The paper has three enthusiastic reviewer with high quality reviews, and one unresponsive reviewer with a short review.

**Justification For Why Not Higher Score:**

N/A

**Justification For Why Not Lower Score:**

This is an important problem, where the community standards are currently shockingly low. We really need this paper, and we really need it to get attention. As the reviewers also illustrate, it is well carried out -- it's an important paper.

---

### Decision · Program_Chairs · 2024-01-16

Accept (oral)